# Uncertainty-Driven Loss for Single Image Super-Resolution

**Qian Ning**[1], **Weisheng Dong**[1]*, **Xin Li**[2], **Jinjian Wu**[1], **Guangming Shi**[1]
[1]School of Artificial Intelligence, Xidian University, Xi'an 710071, China
[2]Lane Dep. of CSEE, West Virginia University, Morgantown WV 26506, USA
`ningqian@stu.xidian.edu.cn`, `{wsdong,jinjian.wu}@mail.xidian.edu.cn`
`xin.li@mail.wvu.edu`, `gmshi@xidian.edu.cn`

## Abstract

In low-level vision such as single image super-resolution (SISR), traditional *MSE* or $\mathcal{L}_1$ loss function treats every pixel equally with the assumption that the importance of all pixels is the same. However, it has been long recognized that texture and edge areas carry more important visual information than smooth areas in photographic images. How to achieve such spatial adaptation in a principled manner has been an open problem in both traditional model-based and modern learning-based approaches toward SISR. In this paper, we propose a new adaptive weighted loss for SISR to train deep networks focusing on challenging situations such as textured and edge pixels with high uncertainty. Specifically, we introduce variance estimation characterizing the uncertainty on a pixel-by-pixel basis into SISR solutions so the targeted pixels in a high-resolution image (mean) and their corresponding uncertainty (variance) can be learned simultaneously. Moreover, uncertainty estimation allows us to leverage conventional wisdom such as sparsity prior for regularizing SISR solutions. Ultimately, pixels with large certainty (e.g., texture and edge pixels) will be prioritized for SISR according to their importance to visual quality. For the first time, we demonstrate that such uncertainty-driven loss can achieve better results than *MSE* or $\mathcal{L}_1$ loss for a wide range of network architectures. Experimental results on three popular SISR networks show that our proposed uncertainty-driven loss has achieved better PSNR performance than traditional loss functions without any increased computation during testing. The code is available at https://see.xidian.edu.cn/faculty/wsdong/Projects/UDL-SR.htm

## 1 Introduction

Single image super-resolution (SISR) aims at reconstructing high-resolution (HR) images from their corresponding degraded low-resolution (LR) images. Since the publication of super-resolution with convolutional neural network (SRCNN) [1], there has been a flurry of works on deep learning-based approaches toward SISR - e.g., EDSR [2], DPDNN [3], RCAN [4], SAN [5], and MoG-DUN [6]. The unifying theme along this line of research appears to be that deeper, bigger, and more complex networks can achieve improved SISR performance by facilitating the reconstruction of high-frequency details such as textures and edges in photographic images. Such improvement has been achieved by novel network architectures (e.g., skip connections [2]), new attention mechanism (e.g., residue channel attention [4]), and closed-loop supervision [7]. Surprisingly, most of these existing methods have adopted *MSE* or $\mathcal{L}_1$ loss to optimize the parameters of networks.

The commonly used practice, such as *MSE* or $\mathcal{L}_1$ loss, treats every pixel equally regardless of whether the pixel is in texture/edge regions or smooth areas. The optimality of such non-adaptive loss function

---

*Corresponding author.

35th Conference on Neural Information Processing Systems (NeurIPS 2021).

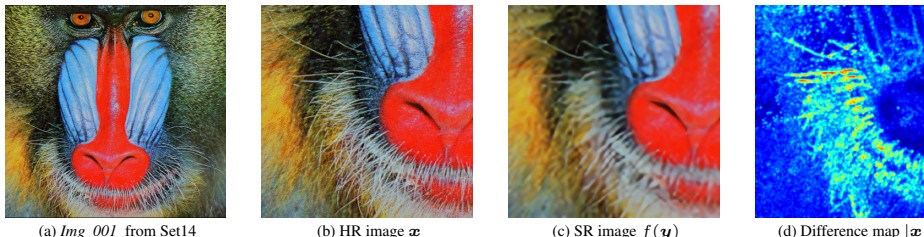

(a) *Img_001* from Set14  (b) HR image $\boldsymbol{x}$  (c) SR image $f(\boldsymbol{y})$  (d) Difference map $|\boldsymbol{x} - f(\boldsymbol{y})|$

Figure 1: Illustration of the difference (d) between HR image (b) and SR image (c) reconstructed by EDSR network [2] on dataset Set14 [8]. The image reconstructed by EDSR network is shown in (c) and (d) shows the absolute difference between the HR image and SR image. **Best viewed in color.**

has been questioned in the literature of SISR calling for the proposition of perceptual loss function (e.g., [9]). From a Bayesian perspective, the assumption underlying the *MSE* or $\mathcal{L}_1$ loss is that each pixel obeys the independent and identically distribution with the same variance. Taking $\mathcal{L}_1$ loss as an example, the likelihood of all pixels in an image can be formulated as

$$p(\boldsymbol{x} \mid \boldsymbol{y}, \boldsymbol{W}) = \prod_{l=1}^{M} c \exp(-\frac{||\boldsymbol{x}^{(l)} - f^{(\boldsymbol{W})}(\boldsymbol{y}^{(l)})||_1}{\sigma}), \tag{1}$$

where $\boldsymbol{x}$ and $\boldsymbol{y}$ denote the pair of HR and LR image, $f^{(\boldsymbol{W})}(\cdot)$ denotes an arbitrary SISR network parameterized by $\boldsymbol{W}$, and $c, \sigma$ denote spatially invariant constants. However, such assumption of stationarity or spatial invariance of image prior model is invalid for photographic images in the real world. For instance, if one compares the ground-truth (HR image) and the SR image reconstructed by EDSR [2] as shown in Fig. 1 (c), it can be observed that texture areas (e.g., hair of baboon) are not restored as good as smooth areas (e.g., nose of baboon). Fig. 1 (d) depicts the absolute difference between the HR image and reconstructed SR image, from which we can observe spatial variation of the difference map. Such observation implies that the uncertainty of texture and edge areas as characterized by the variance is much larger than that in smooth areas. How to address such uncertainty-driven loss for SISR sets up the stage for this paper.

In this paper, we propose a new adaptive weighted loss (uncertainty-driven loss) for SISR by assigning texture and edge areas with higher weights during the training process. Unlike previous work of perceptual loss [9] focusing on characterizing content and style consistency, we target at explicitly estimating the variance field underlying the unknown HR image in the first step, which can be exploited as an auxiliary signal for guiding the SISR solution in the second step. A direct consequence of our two-step learning approach is that it delivers not only higher visual quality but also improved objective performance such as PSNR and SSIM. Moreover, uncertainty estimation perspective allows us to easily incorporate existing models such as Jefferey's prior [10, 11] into the proposed SISR solution. It follows that the network training boils down to two sequential steps in which the variance map is estimated from the first step and serves as the attention signal for the second step. The main technical contributions are summarized as follows.

- Uncertainty modeling and estimation. We propose to cast SISR into a Bayesian estimation framework under which SR image (mean) and uncertainty (variance) are derived simultaneously. Unlike previous works in which pixels with large uncertainty are attenuated for high-level vision tasks, we advocate to prioritize them for low-level vision tasks such as SISR.

- Uncertainty-driven loss (UDL). The estimation of variance map facilitates the training of SISR network by dividing it into two steps. In the first step, an estimating sparsity uncertainty (ESU) loss function was derived from the classical Jeffrey's prior to estimate the variance map. In the second step, the estimated variance map serves as the guidance signal leading to adaptive weighted loss named uncertainty-driven loss $\mathcal{L}_{\text{UDL}}$.

- Universality of UDL. The proposed uncertainty loss can easily be employed in any existing SISR network to improve performance and do not increase any additional computation cost during testing.

- Experimental results on three different baseline networks show that our proposed uncertainty-driven loss has achieved better PSNR performance than traditional *MSE* or $\mathcal{L}_1$ loss.

## 2 Related Work

### 2.1 Uncertainty in Deep Learning

Many works [12–14] have introduced uncertainty into the regression with input-dependent noises problems, and studied the nature and behavior of uncertainty for a long time. More recently, modeling uncertainty in deep learning have improved the performance and robustness of deep networks in many computer vision tasks [15–17] such as image classification [18], image segmentation [15, 16], and face recognition [17, 19]. The uncertainty in deep learning can be roughly divided into two categories [20]. Epistemic/model uncertainty describes how much the model is uncertain about its predictions. Another type is aleatoric/data uncertainty which refers to noise inherent in observation data. In [15], they presented a Bayesian deep learning framework combining aleatoric uncertainty with epistemic uncertainty for per-pixel semantic segmentation and depth regression tasks. Chang *et al.*[17] investigated the data uncertainty with estimated mean and variance in face recognition. Those uncertainty-based loss function proposed by those works [15–17] can be summarized as

$$\mathcal{L} = \frac{1}{N} \sum_{i=1}^{N} \frac{||\boldsymbol{x}_i - f(\boldsymbol{y}_i)||_2}{2\,\sigma_i^2} + \frac{1}{2} \ln \sigma_i^2, \tag{2}$$

where $f(\boldsymbol{y}_i)$ and $\sigma_i^2$ denote the learned mean and variance respectively. Using above loss function indeed improved their robustness to noisy data. In those tasks, the pixels with high uncertainty were regarded as unreliable pixels which would bear loss attenuation. On the contrary, in SISR tasks, the pixels with high uncertainty (e.g., complex texture or edge areas) should be prioritized since those regions visually more important than pixels in smooth areas. That can explain why applying above loss into SISR directly leads performance decline.

### 2.2 Modeling Uncertainty for SISR

To the best of our knowledge, only two works [21, 22] have studied the behavior of uncertainty for SISR in the open literature. [22] used batch-normalization uncertainty to analyze SISR uncertainty, improving the robustness of the network against adversarial attack. The most recent advance related to our work is Gradient Rescaling Attention Model (GRAM) [21], which analyses the effect of aleatoric/data uncertainty on SISR reconstruction. By decreasing the loss attenuation of large variance pixels, GRAM achieves better results than applying above uncertainty loss into SISR directly. However, GRAM [21] loss remains attenuated when the variance of pixels is high, which contradicts the intuition of prioritizing texture and edge pixels. Thus, GRAM [21] is still inferior to baseline methods since the proposed method fails to prioritize the pixels of large variance. Different from GRAM, we propose a novel uncertainty-driven loss (UDL) to enforce the network concentrating more on the pixels with large variance aiming at better reconstruction of texture and edge regions. By quantifying the uncertainty in SISR under deep Bayesian framework, our proposed method has achieved better results than baseline methods.

## 3 Methodology

Unlike traditional *MSE* or $\mathcal{L}_1$ loss treating every pixel equally, the proposed new adaptive weighted loss for SISR aims at prioritizing texture and edge pixels that are visually more important than pixels in smooth areas. Toward this objective, we first introduce an approach of estimating intermediate results of SR image (mean) and uncertainty (variance) simultaneously in SISR. Then, with Jeffrey's prior term, a regularized approach of estimating sparse uncertainty is proposed for more accurate uncertainty estimation. An important new insight brought by this paper is that *unlike high-level vision tasks where pixels with large uncertainty are assigned lower weights to attenuate their impact [15], one should prioritize these pixels in low-level vision tasks such as SISR*. Such observation implies that the attenuation of weighting coefficients in loss function needs to be properly translated into the attention mechanism given the specific vision problem as the context.

In previous study [15], it has been shown that explicitly representing aleatoric uncertainty can lead to performance and robustness improvement to noise data in high-level vision tasks such as image segmentation. Such improvement can be explained away by attenuating the weights of pixels with

large uncertainty. However, attenuation has to go the *opposite* direction in low-level vision tasks such as SISR - i.e., larger weights should be assigned to the pixels with high uncertainty (e.g., texture and edge pixels) because they are visually more important than pixels in smooth regions. It should be noted that existing work such as gradient rescaling strategy in GRAM [21] fails to recognize such difference and does not prioritize pixels with high uncertainty. In this paper, we propose a new adaptive weighted loss named uncertainty-driven loss (UDL) for properly turning attenuation into attention for SISR.

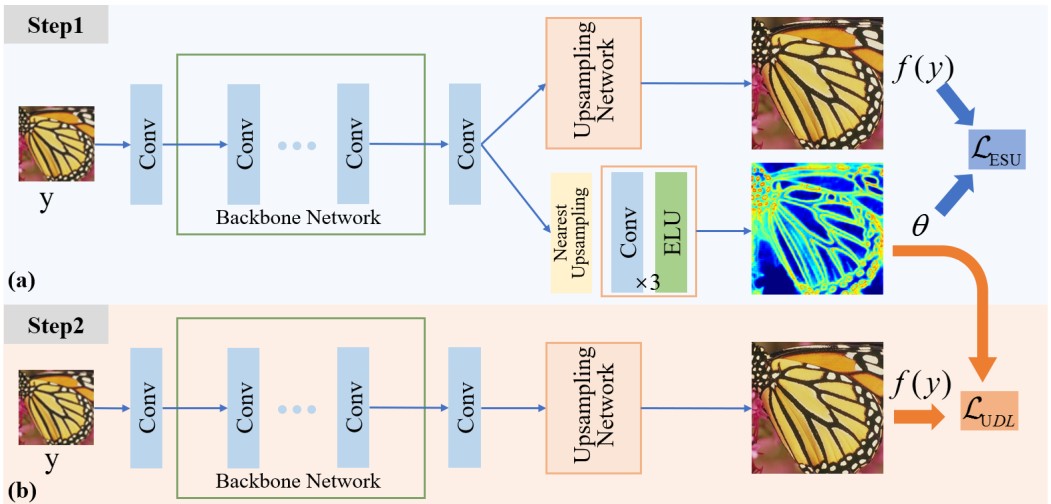

Figure 2: The overview of training SISR network with proposed $\mathcal{L}_{\text{UDL}}$ loss. The whole training process can divided into two steps; the first step estimates the uncertainty $\boldsymbol{\theta}$ precisely and the second step generates the final mean value $f(\boldsymbol{y})$. In step1 shown in (a), the mean value $f(\boldsymbol{y})$ and variance $\boldsymbol{\theta}$ are pretrained by $\mathcal{L}_{\text{ESU}}$ loss. During step2, as shown in (b), the mean value $f(\boldsymbol{y})$ network is trained by $\mathcal{L}_{\text{UDL}}$ loss, while the network of inferring variance $\boldsymbol{\theta}$ is fixed. Note that the mean value $f(\boldsymbol{y})$ network of step2 starts training from the pretrained network of step1. The Nearest Upsampling denotes interpolation operator.

### 3.1 Estimating Uncertainty (EU) in SISR.

As discussed in [15], there are two classes of uncertainty in Bayesian modeling: aleatoric uncertainty capturing noise inherent in observation data and epistemic uncertainty accounting for uncertainty of model about its predictions. We opt to study the former (aleatoric uncertainty) and explore its application into SISR by designing new uncertainty-driven loss (UDL) functions in this paper. In order to better quantify aleatoric uncertainty in SISR, we use $\boldsymbol{y}_i, \boldsymbol{x}_i$ to denote the low-resolution (LR) image and the corresponding high-resolution (HR) image respectively. Let $f(\cdot)$ denotes an arbitrary SISR network and the aleatoric uncertainty can be denoted by an additive term $\boldsymbol{\theta}_i$. This way, the overall observation model can be formulated as

$$\boldsymbol{x}_i = f(\boldsymbol{y}_i) + \epsilon\,\boldsymbol{\theta}_i, \tag{3}$$

where $\epsilon$ represents the Laplace distribution with zero-mean and unit-variance. Existing deep-learning based SISR methods target at training a network to learn the SR image (mean) $f(\boldsymbol{y}_i)$ only. To more accurately characterize aleatoric uncertainty for SISR, we propose to estimate not only the SR image (mean) $f(\boldsymbol{y}_i)$ but also the uncertainty (variance) $\boldsymbol{\theta}_i$ simultaneously.

For a given LR image $\boldsymbol{y}_i$ and corresponding HR image $\boldsymbol{x}_i$, a Laplace distribution [2] is assumed for characterizing the likelihood function by

$$p(\boldsymbol{x}_i, \boldsymbol{\theta}_i | \boldsymbol{y}_i) = \frac{1}{2\,\boldsymbol{\theta}_i} \exp(-\frac{||\boldsymbol{x}_i - f(\boldsymbol{y}_i)||_1}{\boldsymbol{\theta}_i}), \tag{4}$$

where $f(\boldsymbol{y}_i)$ and $\boldsymbol{\theta}_i$ denote the SR image (mean) and the uncertainty (variance) which are learned by deep neural networks (DNNs) respectively. Then, the log likelihood can be formulated as follows,

$$\ln p(\boldsymbol{x}_i, \boldsymbol{\theta}_i | \boldsymbol{y}_i) = -\frac{||\boldsymbol{x}_i - f(\boldsymbol{y}_i)||_1}{\boldsymbol{\theta}_i} - \ln \boldsymbol{\theta}_i - \ln 2 \tag{5}$$

---

[2]The most commonly used loss function in SISR is $L_1$ loss which refers to Laplace distribution.

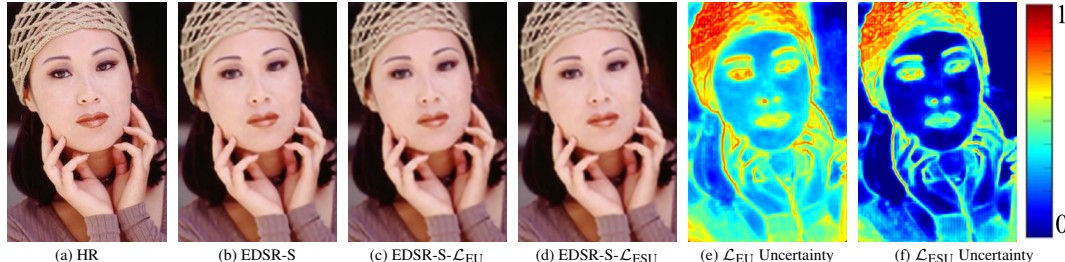

| (a) HR | (b) EDSR-S | (c) EDSR-S-$\mathcal{L}_{EU}$ | (d) EDSR-S-$\mathcal{L}_{ESU}$ | (e) $\mathcal{L}_{EU}$ Uncertainty | (f) $\mathcal{L}_{ESU}$ Uncertainty |

Figure 3: SISR visual quality comparisons of EDSR-S [2] with different loss function on 'Img_005' from Set5 [23] (bicubic-downsampling $\times 4$). **Best viewed in color.**

For numerical stability, we train the networks to estimate log variance $\boldsymbol{s}_i = \ln \boldsymbol{\theta}_i$ as shown in Fig. 2 (a). At last, the maximum likelihood estimation of (5) can be reformulated as the minimization of following loss function for estimating uncertainty (EU) in SISR.

$$\mathcal{L}_{EU} = \frac{1}{N} \sum_{i=1}^{N} \exp(-\boldsymbol{s}_i) \, ||\boldsymbol{x}_i - f(\boldsymbol{y}_i)||_1 + \boldsymbol{s}_i \tag{6}$$

**Jeffrey's Prior for Estimating Sparse Uncertainty (ESU) in SISR.** The loss function $\mathcal{L}_{EU}$ includes two terms; the first one is associated with fidelity term and the second one prevents the network from predicting infinite uncertainty for all pixels. Those two terms reach equilibrium but there is no prior that imposed on the uncertainty estimation. Therefore, based on the observation that the uncertainty is sparse in view of the whole image as shown in Fig. 2, we propose to impose Jeffrey's prior [10] $p(\boldsymbol{w}) \propto \frac{1}{\boldsymbol{w}}$ on uncertainty $\boldsymbol{\theta}_i$, which can be expressed as

$$p(\boldsymbol{x}_i, \boldsymbol{\theta}_i | \boldsymbol{y}_i) = p(\boldsymbol{x}_i | \boldsymbol{y}_i, \boldsymbol{\theta}_i) p(\boldsymbol{\theta}_i) \propto \frac{1}{2\,\boldsymbol{\theta}_i} \exp(-\frac{||\boldsymbol{x}_i - f(\boldsymbol{y}_i)||_1}{\boldsymbol{\theta}_i}) \frac{1}{\boldsymbol{\theta}_i} = \frac{1}{2\,\boldsymbol{\theta}_i^2} \exp(-\frac{||\boldsymbol{x}_i - f(\boldsymbol{y}_i)||_1}{\boldsymbol{\theta}_i}) \tag{7}$$

Then the log likelihood and loss function can be separately formulated as follows,

$$\ln p(\boldsymbol{x}_i | \boldsymbol{y}_i) = -\frac{||\boldsymbol{x}_i - f(\boldsymbol{y}_i)||_1}{\boldsymbol{\theta}_i} - 2 \ln \boldsymbol{\theta}_i - \ln 2 \tag{8}$$

$$\mathcal{L}_{ESU} = \frac{1}{N} \sum_{i=1}^{N} \exp(-\boldsymbol{s}_i) \, ||\boldsymbol{x}_i - f(\boldsymbol{y}_i)||_1 + 2\,\boldsymbol{s}_i \tag{9}$$

**The limitations of $\mathcal{L}_{\mathbf{EU}}$ and $\mathcal{L}_{\mathbf{ESU}}$ loss.** Applying $\mathcal{L}_{EU}$ and $\mathcal{L}_{ESU}$ loss leads to more accurate estimation of uncertainty (variance field), but counter-intuitively, they do not directly improve the performance of SISR. We have conducted experiments comparing those three different loss functions to verify the above claim. As shown in Tab. 1, the average PSNR and SSIM results of $\mathcal{L}_{ESU}$ and $\mathcal{L}_{EU}$ are notably lower than the original results. The reason behind this observation is that both $\mathcal{L}_{EU}$ and $\mathcal{L}_{ESU}$ loss functions have incorporated the variance term ($\boldsymbol{\theta}_i$) into the divisor of the absolute difference term. Consequently, a pixel with a large variance will be penalized after the division and has less impact on the overall loss function. Note that such attenuation of pixels with large uncertainty is preferred for high-level vision tasks, as demonstrated in previous works [15–17] on image classification [18], image segmentation [15, 16], and face recognition [17, 19].

Low-level vision tasks such as SISR are much different. As shown in Fig. 1, pixels with large uncertainty carry visually important information such as textured and edges. They need to be prioritized (opposite to attenuation) and given larger instead of smaller weights. To verify such claim, we have presented a simple example comparing the visual results between $\mathcal{L}_{EU}$ and $\mathcal{L}_{ESU}$ as shown in Fig. 3. It can be seen that the uncertainty captured by $\mathcal{L}_{ESU}$ loss is better than $\mathcal{L}_{EU}$ loss. The improvement of $\mathcal{L}_{ESU}$ in Eq. (9) over $\mathcal{L}_{EU}$ in Eq. (6) is attributed to the prioritization of pixels with large uncertainty ($\boldsymbol{s}_i$ values). Fig. 3 (f) clearly demonstrate superiority of exploiting the sparsity constraint with the uncertainty estimation.

Table 1: Average PSNR and SSIM results for **BI** degradation on five datasets for investigating three different loss. The best performance is shown in **bold**. We record the results in $1.2 \times 10^5$ iterations.

| Base Model | Scale | Loss | Set5 [23] | | Set14 [8] | | BSD100 [24] | | Urban100 [25] | | Manga109 [26] | |
|---|---|---|---|---|---|---|---|---|---|---|---|---|
| | | | PSNR | SSIM | PSNR | SSIM | PSNR | SSIM | PSNR | SSIM | PSNR | SSIM |
| EDSR-S[2] | ×4 | Original | **30.93** | **0.8740** | **27.80** | **0.7627** | **27.05** | **0.7190** | **24.71** | **0.7351** | **28.14** | **0.8693** |
| | | $\mathcal{L}_{EU}$ | 30.19 | 0.8627 | 27.29 | 0.7538 | 26.78 | 0.7120 | 24.21 | 0.7179 | 26.78 | 0.8481 |
| | | $\mathcal{L}_{ESU}$ | 30.31 | 0.8637 | 27.39 | 0.7543 | 26.83 | 0.7124 | 24.27 | 0.7192 | 26.92 | 0.8496 |

## 3.2 Uncertainty-Driven Loss (UDL) for SISR

Improvement of $\mathcal{L}_{ESU}$ over $\mathcal{L}_{EU}$ inspired us to go one step further. To better prioritize pixels with large uncertainty, we propose a new adaptive weighted loss named uncertainty-driven loss (UDL) for SISR. Unlike $\mathcal{L}_{ESU}$ loss putting a larger weight to the second term than $\mathcal{L}_{EU}$, we suggest that the first term can also be modified to directly associate the aleatoric/data uncertainty of $f(\boldsymbol{y}_i)$. That is, instead of using $exp(-\boldsymbol{s}_i)$ to attenuate the importance of pixels with large uncertainty, we need to use a monotonically increasing function to prioritize them. Linear scaling would be a natural option, which leads to the following loss function

$$\mathcal{L}_{UDL} = \frac{1}{N} \sum_{i=1}^{N} \hat{\boldsymbol{s}}_i \, \|\boldsymbol{x}_i - f(\boldsymbol{y}_i)\|_1, \tag{10}$$

where $\hat{\boldsymbol{s}}_i = \boldsymbol{s}_i - \min(\boldsymbol{s}_i)$ is a non-negative linear scaling function. To prevent uncertainty value from degenerating into zeros, the result of uncertainty estimation network in the first step will be passed to the second step as the attention signal ($\boldsymbol{s} = \ln \boldsymbol{\theta}$), as shown in Fig. 2. By leveraging the log variance to represent the challenging and cumbersome pixels with higher uncertainty, we propose a new weighted loss named uncertainty-driven loss $\mathcal{L}_{UDL}$. In $\mathcal{L}_{UDL}$ loss, texture and edge pixels with higher uncertainty tend to have larger weights than those in smooth regions. In summary, the uncertainty estimation $\boldsymbol{\theta}$ serves as the bridge connecting two steps: it is the output of the first step; but passed on to the second step as the guidance required for calculating $\mathcal{L}_{UDL}$ loss.

## 3.3 Two-step Training of Dual Networks

As shown in Fig. 2, the whole training process can be divided into two steps; the first step estimates the uncertainty $\boldsymbol{\theta}$ precisely and the second step generates the final mean value $f(\boldsymbol{y})$ with the aid from the estimated uncertainty $\boldsymbol{\theta}$ from step1. More specifically, the mean value $f(\boldsymbol{y})$ and variance $\theta$ are pre-trained by $\mathcal{L}_{ESU}$ loss during step1 as shown in Fig. 2 (a). After the uncertainty has been estimated, the mean value $f(\boldsymbol{y})$ network is trained by $\mathcal{L}_{UDL}$ loss with variance $\theta$ as shown in Fig. 2 (b), while the network of inferring variance $\boldsymbol{\theta}$ is fixed.

Note that the mean value $f(\boldsymbol{y})$ network of step2 starts training from the pre-trained network of step1. Such partial parameter sharing is a salient property of our proposed dual networks with parallel symmetric attention [27]. In theory, we can extend the two-step training into multiple-step training by alternating between the estimation of uncertainty (variance $\boldsymbol{\theta}$) and mean value $f(\boldsymbol{y})$. Conceptually, an improved estimation of unknown HR image can leads to an improved estimation of aleatoric uncertainty and vice versa. This line of reasoning will lead to the pursuit of a deep equilibrium model [28] for SISR; but it is beyond the scope of this paper.

## 3.4 Discussions: Why UDL Outperforms GRAM?

To the best of our knowledge, only one work GRAM [21] has studied data uncertainty in SISR, which is the most related to our work. We will discuss connections and differences between proposed UDL and GRAM [21] here. First, both GRAM [21] and our work has found out that applying the traditional uncertainty loss designed for high-level computer vision tasks into SISR task directly results in performance decline. For high-level computer vision tasks, the pixels with higher uncertainty indicates less confidence in final inference, which needs loss attenuation. However, for SISR tasks, the pixels with higher uncertainty (e.g., texture and edge pixels) should be prioritized with larger weights because they are visually more important than pixels in smooth regions. To solve this problem, GRAM [21] proposes to use uncertainty to generate an attention mask that decreases loss attenuation. However, GRAM [21] loss still is attenuated when the variance of pixels is high. Thus, GRAM [21] is still inferior to baseline method since it still does not prioritize pixels with high uncertainty.

Different from GRAM, we propose an uncertainty-driven loss to assign the pixels with high variance more weight to prioritize them. Besides, modeling uncertainty under Bayesian framework allows us to leverage sparsity prior for a more precise estimation of uncertainty. Ultimately, our proposed method consists of those two technical contributions that achieve better results than baseline methods and outperform GRAM.

# 4 Experiments

## 4.1 Experimental Settings

**Datasets and Metrics.** 800 high-quality (2K resolution) images from the DIV2K dataset [29] have been used for training. Following EDSR [2], five standard benchmark datasets: Set5 [23], Set14 [8], BSD100[24], Urban100 [25], Manga109 [26] are used for testing. Performance evaluation in terms of of PSNR and SSIM [30] metrics is conducted on the luminance (Y) channel only.

**Training Setting.** We randomly select 16 RGB LR patches sized by $48 \times 48$ as the inputs. The image patches are randomly rotated by $90°, 180°, 270°$ and flipped horizontally. The ADAM algorithm [31] with $\beta_1 = 0.9, \beta_2 = 0.999, \epsilon = 10^{-8}$ is adopted to optimize the network. The initial learning rate is $10^{-4}$ and decreases by half for every $2 \times 10^5$ minibatch updates.

**Degradation models.** To demonstrate the effectiveness of our proposed uncertainty-driven loss in varying degradation scenarios, we have designed the following experiments with two different degradation models. Let **BI** denotes bicubic downsampling. The second one is **BD** which uses Gaussian blur followed by nearest downsampling to generate LR images. Specifically, we apply $11 \times 11$ sized Gaussian kernel with a standard deviation $1.6$ for blurring in our experiments.

**SISR Networks.** We choose three different networks to verify the effectiveness of proposed $\mathcal{L}_{UDL}$ loss. The first one is EDSR-S or called baseline network in [2]. EDSR-S [2] mainly consists of 16 Resblock with 64 channels, having $1.5M$ parameters. The second one is DPDNN[3] where denoiser network is U-net under model-guided framework. The last one is a big network EDSR[2], consisting of 32 Resblock with 256 channels, having $43M$ parameters. The analysis of training cost can be found in our supplementary material.

## 4.2 Ablation Study

Table 2: Average PSNR and SSIM results for **BI** degradation on five datasets for investigating three different loss. The best performance is shown in **bold**. We record the results in $4 \times 10^5$ iterations.

| Base Model | Scale | Loss | Set5 [23] | | Set14 [8] | | BSD100 [24] | | Urban100 [25] | | Manga109 [26] | |
|---|---|---|---|---|---|---|---|---|---|---|---|---|
| | | | PSNR | SSIM | PSNR | SSIM | PSNR | SSIM | PSNR | SSIM | PSNR | SSIM |
| | | Original | 31.61 | 0.8862 | 28.22 | 0.7721 | 27.30 | 0.7271 | 25.25 | 0.7575 | 29.31 | 0.8907 |
| EDSR-S[2] | $\times 4$ | $\mathcal{L}_{EU}+\mathcal{L}_{UDL}$ | 31.83 | 0.8895 | 28.33 | 0.7754 | 27.37 | 0.7297 | 25.49 | 0.7665 | 29.70 | 0.8959 |
| | | $\mathcal{L}_{ESU}+\mathcal{L}_{UDL}$ | **31.90** | **0.8897** | **28.37** | **0.7755** | **27.40** | **0.7301** | **25.54** | **0.7671** | **29.77** | **0.8967** |

To further verify the effectiveness of sparse uncertainty estimation at step1, we have conducted an ablation study to compare the final PSNR/SSIM results of $\mathcal{L}_{UDL}$ with $\mathcal{L}_{EU}$ or with $\mathcal{L}_{ESU}$ at step1. In our ablation study, we have used $\times 4$ bicubic down-sampling degradation on five frequently-used benchmark datasets with EDSR-S backbone[2]. As shown in Tab. 2, both $\mathcal{L}_{EU}+\mathcal{L}_{UDL}$ and $\mathcal{L}_{ESU}+\mathcal{L}_{UDL}$ loss have achieved better performance than original loss. Besides, $\mathcal{L}_{ESU}+\mathcal{L}_{UDL}$ loss obtains better results than $\mathcal{L}_{EU}+\mathcal{L}_{UDL}$ due to more accurate uncertainty estimation as shown in Fig. 3 (e) and (f).

## 4.3 Analysis of Different Weighted Loss

There are many different weighted loss guided by different weight maps, such as *Error_map, Gradient_map* which can also reveal the challenging pixels. We have conducted experiments with a weighted loss function where the weight is a pixel-wise gradient or *Error_map*. The PSNR results of five benchmark datasets for investigating the influence of different weighted loss functions can be summarized in Tab. 3.

The *HR_gradient_map* and *LR_gradien_map* denote calculating gradient map from high-resolution (ground truth) images and low-resolution images respectively. The calculation of gradient can be formulated as

$$V(i,j) = I(i+1,j) - I(i,j), H(i,j) = I(i,j+1) - I(i,j), G(i,j) = ||(V(i,j), H(i,j)||_2, \quad (11)$$

Table 3: Average PSNR and $\Delta$ PSNR results with **BI** degradation on five datasets for investigating the influence of different weighted loss functions. The best performance is shown in **bold**.

| Weighted loss | Set5 | $\Delta$ | Set14 | $\Delta$ | BSD100 | $\Delta$ | Urban100 | $\Delta$ | Manga109 | $\Delta$ |
|---|---|---|---|---|---|---|---|---|---|---|
| Baseline | 31.61 | 0.00 | 28.22 | 0.00 | 27.30 | 0.00 | 25.25 | 0.00 | 29.31 | 0.00 |
| Uncertainty(Ours) | **31.90** | **0.29↑** | **28.37** | **0.15↑** | **27.40** | **0.10↑** | **25.54** | **0.29↑** | **29.77** | **0.46↑** |
| Error_map | 31.77 | 0.16↑ | 28.30 | 0.08↑ | 27.35 | 0.05↑ | 25.40 | 0.15↑ | 29.57 | 0.26↑ |
| HR_gradient_map | 31.68 | 0.07↑ | 28.27 | 0.05↑ | 27.35 | 0.05↑ | 25.42 | 0.17↑ | 29.45 | 0.14↑ |
| LR_gradient_map | 31.69 | 0.08↑ | 28.29 | 0.07↑ | 27.35 | 0.05↑ | 25.38 | 0.13↑ | 29.50 | 0.19↑ |

where $I$ denotes pixels value and $i, j$ denotes position of pixels. Note that we adjust the scaling functions of *Error_map, HR_gradient_map and LR_gradient_map* to get the best performance.

From the Tab. 3, one can be observed that other weighted loss functions can indeed improve the PSNR results, but only to certain degrees. Comparing four different weight maps, our proposed uncertainty weighted loss function can bring the biggest improvement. Although the *Error_map* can represent the variance of a single pixel, the *Error_map* lacks semantic information or local information to capture a more precise estimation of variance comparing uncertainty. With regard to the gradient map of HR or LR images, those gradient maps only well match the edges of images and have a certain correlation to variance. Comparing the visual results of *Error_map, HR_gradient_map* and *LR_gradient_map* with *uncertainty map*, those maps only detect edges of images and fail reflecting complex texture details which are important to final reconstruction performance. Therefore, uncertainty-weighted loss can is still valuable for achieving the best performance among other weighted maps.

### 4.4 Analysis of Different Scaling Functions

We have conducted experiments with several various monotonically increasing functions (including linear and non-linear) and the results can be summarized in Tab. 4.

Table 4: Average PSNR and $\Delta$ PSNR results with **BI** degradation on five datasets for investigating the influence of different scaling functions. The best performance are shown in **bold**.

| Scaling functions | Set5 | $\Delta$ | Set14 | $\Delta$ | BSD100 | $\Delta$ | Urban100 | $\Delta$ | Manga109 | $\Delta$ |
|---|---|---|---|---|---|---|---|---|---|---|
| Baseline | 31.61 | 0.00 | 28.22 | 0.00 | 27.30 | 0.00 | 25.25 | 0.00 | 29.31 | 0.00 |
| $s - min(s)$ | **31.90** | **0.29↑** | 28.37 | 0.15↑ | 27.40 | 0.10↑ | 25.54 | 0.29↑ | **29.77** | **0.46↑** |
| $exp(s)$ | 31.80 | 0.19↑ | 28.34 | 0.12↑ | 27.40 | 0.10↑ | 25.53 | 0.28↑ | 29.66 | 0.35↑ |
| $exp(s)^{(1/2)}$ | 31.86 | 0.25↑ | 28.36 | 0.14↑ | 27.41 | 0.11↑ | 25.55 | 0.30↑ | 29.71 | 0.40↑ |
| $log(s) - min(log(s))$ | 31.89 | 0.28↑ | **28.39** | **0.17↑** | **27.42** | **0.12↑** | **25.57** | **0.32↑** | 29.74 | 0.43↑ |

The best and second-best performances are shown in **bold**. Overall, four various monotonically increasing functions have achieved better results than the baseline method. The best two scaling functions are linear scaling and log scaling with a slight difference as shown in the above table. Since the linear scaling function achieves a comparable performance with low computational cost, we advocate this choice in this paper.

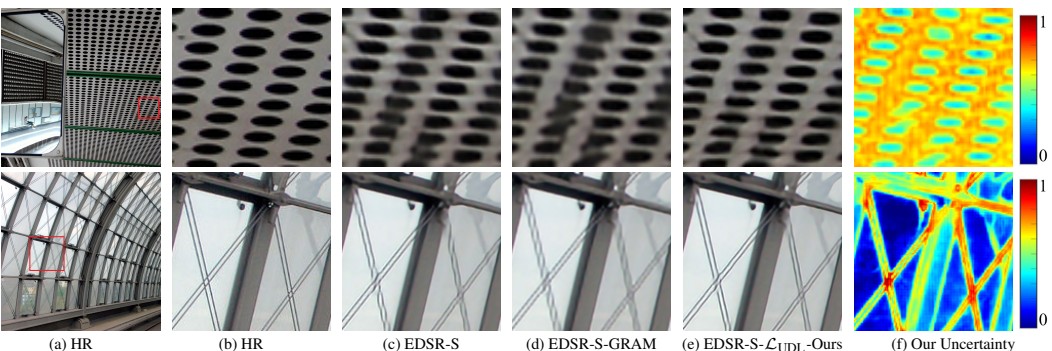

(a) HR    (b) HR    (c) EDSR-S    (d) EDSR-S-GRAM    (e) EDSR-S-$\mathcal{L}_{UDL}$-Ours    (f) Our Uncertainty

Figure 4: SISR visual quality comparisons of EDSR-S [2] with different loss function on 'Img_004' and 'Img_016' from Urban100 [25] (bicubic-downsampling $\times 4$). **Best viewed in color.**

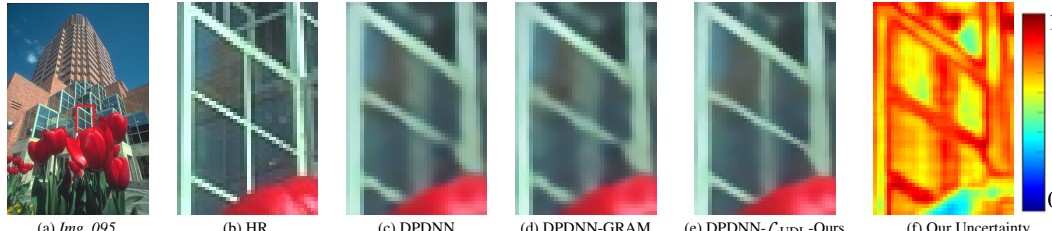

| (a) *Img_095* | (b) HR | (c) DPDNN | (d) DPDNN-GRAM | (e) DPDNN-$\mathcal{L}_{\text{UDL}}$-Ours | (f) Our Uncertainty |

Figure 5: SISR visual quality comparisons of DPDNN [3] with different loss function on 'Img_095' from BSD100 [24] (bicubic-downsampling ×4). **Best viewed in color.**

Table 5: Average PSNR and SSIM results for **BI** degradation on five benchmark datasets. The best performance is shown in **bold**. Note that $\mathcal{L}_{\text{UDL}}$-Ours denotes adopting $\mathcal{L}_{\text{ESU}}$ at step1 and $\mathcal{L}_{\text{UDL}}$ at step2 for simplicity.

| Base Model | Scale | Loss | Set5 [23] | | Set14 [8] | | BSD100 [24] | | Urban100 [25] | | Manga109 [26] | |
|---|---|---|---|---|---|---|---|---|---|---|---|---|
| | | | PSNR | SSIM | PSNR | SSIM | PSNR | SSIM | PSNR | SSIM | PSNR | SSIM |
| EDSR-S [2] | ×2 | Original | 37.66 | 0.8594 | 33.22 | 0.9146 | 31.95 | 0.8969 | 30.71 | 0.9205 | 37.79 | 0.9752 |
| | | GRAM [21] | 37.48 | 0.9589 | 32.99 | 0.9126 | 31.76 | 0.8946 | 30.11 | 0.9134 | 37.38 | 0.9739 |
| | | $\mathcal{L}_{\text{UDL}}$-Ours | **37.95** | **0.9604** | **33.50** | **0.9165** | **32.13** | **0.8991** | **31.54** | **0.9304** | **38.38** | **0.9767** |
| DPDNN [3] | ×2 | Original | 37.75 | 0.9600 | 33.30 | 0.9150 | 32.09 | 0.8990 | 31.50 | 0.9220 | - | - |
| | | GRAM [21] | 37.74 | 0.9597 | 33.27 | 0.9148 | 31.98 | 0.8973 | 30.97 | 0.9238 | 38.14 | 0.9758 |
| | | $\mathcal{L}_{\text{UDL}}$-Ours | **38.00** | **0.9605** | **33.63** | **0.9176** | **32.16** | **0.8995** | **31.72** | **0.9331** | **38.55** | **0.9769** |
| EDSR [2] | ×2 | Original | 38.11 | 0.9602 | 33.92 | 0.9195 | 32.32 | 0.9013 | 32.93 | 0.9351 | 39.10 | 0.9773 |
| | | GRAM [21] | 37.87 | 0.9604 | 33.43 | 0.9164 | 32.08 | 0.8990 | 31.46 | 0.9301 | 37.91 | 0.9765 |
| | | $\mathcal{L}_{\text{UDL}}$-Ours | **38.29** | **0.9615** | **34.14** | **0.9236** | **32.40** | **0.9027** | **32.99** | **0.9446** | **39.53** | **0.9787** |
| EDSR-S [2] | ×3 | Original | 33.90 | 0.9231 | 29.95 | 0.8352 | 28.85 | 0.7996 | 27.30 | 0.8344 | 32.52 | 0.9369 |
| | | GRAM [21] | 33.27 | 0.9178 | 29.60 | 0.8298 | 28.60 | 0.7936 | 26.52 | 0.8142 | 31.14 | 0.9258 |
| | | $\mathcal{L}_{\text{UDL}}$-Ours | **34.15** | **0.9251** | **30.15** | **0.8388** | **28.99** | **0.8021** | **27.72** | **0.8430** | **32.97** | **0.9406** |
| DPDNN [3] | ×3 | Original | 33.93 | 0.9240 | 30.02 | 0.8360 | 29.00 | 0.8010 | 27.61 | 0.8420 | - | - |
| | | GRAM [21] | 33.92 | 0.9241 | 30.00 | 0.8362 | 28.86 | 0.8000 | 27.37 | 0.8353 | 32.41 | 0.9373 |
| | | $\mathcal{L}_{\text{UDL}}$-Ours | **34.30** | **0.9267** | **30.31** | **0.8419** | **29.10** | **0.8047** | **28.02** | **0.8505** | **33.27** | **0.9435** |
| EDSR [2] | ×3 | Original | 34.65 | 0.9280 | 30.52 | 0.8462 | 29.25 | 0.8093 | 28.80 | 0.8653 | 34.17 | 0.9476 |
| | | GRAM [21] | 34.34 | 0.9270 | 30.28 | 0.8412 | 29.07 | 0.8044 | 27.98 | 0.8789 | 33.32 | 0.9432 |
| | | $\mathcal{L}_{\text{UDL}}$-Ours | **34.83** | **0.9312** | **30.69** | **0.8497** | **29.28** | **0.8109** | **28.99** | **0.8697** | **34.63** | **0.9502** |
| EDSR-S [2] | ×4 | Original | 31.61 | 0.8862 | 28.22 | 0.7721 | 27.30 | 0.7271 | 25.25 | 0.7575 | 29.31 | 0.8907 |
| | | GRAM [21] | 31.08 | 0.8787 | 27.89 | 0.7670 | 27.12 | 0.7229 | 24.81 | 0.7429 | 28.18 | 0.8762 |
| | | $\mathcal{L}_{\text{UDL}}$-Ours | **31.90** | **0.8897** | **28.37** | **0.7755** | **27.40** | **0.7301** | **25.54** | **0.7671** | **29.77** | **0.8967** |
| DPDNN [3] | ×4 | Original | 31.72 | 0.8890 | 28.28 | 0.7730 | 27.44 | 0.7290 | 25.53 | 0.7680 | - | - |
| | | GRAM [21] | 31.89 | 0.8913 | 28.37 | 0.7772 | 27.41 | 0.7314 | 25.63 | 0.7708 | 29.70 | 0.9003 |
| | | $\mathcal{L}_{\text{UDL}}$-Ours | **32.20** | **0.8944** | **28.60** | **0.7819** | **27.56** | **0.7356** | **26.09** | **0.7862** | **30.38** | **0.9082** |
| EDSR [2] | ×4 | Original | 32.46 | 0.8968 | 28.80 | 0.7876 | 27.71 | 0.7420 | 26.64 | 0.8033 | 31.02 | 0.9148 |
| | | GRAM [21] | 32.32 | 0.8971 | 28.73 | 0.7858 | 27.66 | 0.7395 | 26.35 | 0.7955 | 30.73 | 0.9125 |
| | | $\mathcal{L}_{\text{UDL}}$-Ours | **32.59** | **0.8998** | **28.87** | **0.7889** | **27.78** | **0.7431** | **26.75** | **0.8054** | **31.24** | **0.9167** |

## 4.5 Results with BI Degradation Model

For bicubic downsampling (BI), we have compared proposed $\mathcal{L}_{\text{UDL}}$ loss function with GRAM [21] and original loss functions such as *MSE* or $\mathcal{L}_1$ on three different SISR networks. The average PSNR and SSIM results in Tab. 5 are cited from corresponding papers or retrained from officially released code. It is easy to see that our proposed $\mathcal{L}_{\text{UDL}}$ loss function is superior to GRAM [21] and original

Table 6: Average PSNR and SSIM results for **BD** degradation on five benchmark datasets. The best performance is shown in **bold**. Note that $\mathcal{L}_{\text{UDL}}$-Ours denotes adopting $\mathcal{L}_{\text{ESU}}$ at step1 and $\mathcal{L}_{\text{UDL}}$ at step2 for simplicity.

| Base Model | Scale | Loss | Set5 [23] | | Set14 [8] | | BSD100 [24] | | Urban100 [25] | | Manga109 [26] | |
|---|---|---|---|---|---|---|---|---|---|---|---|---|
| | | | PSNR | SSIM | PSNR | SSIM | PSNR | SSIM | PSNR | SSIM | PSNR | SSIM |
| EDSR-S [2] | ×4 | Original | 31.70 | 0.8903 | 28.37 | 0.7778 | 27.37 | 0.7320 | 25.77 | 0.7789 | 29.83 | 0.9014 |
| | | GRAM [21] | 30.98 | 0.8791 | 27.85 | 0.7667 | 27.05 | 0.7225 | 24.79 | 0.7452 | 28.12 | 0.8773 |
| | | $\mathcal{L}_{\text{UDL}}$-Ours | **31.97** | **0.8927** | **28.45** | **0.7793** | **27.41** | **0.7321** | **25.95** | **0.7842** | **30.18** | **0.9053** |
| DPDNN [3] | ×4 | Original | 31.86 | 0.8923 | 28.38 | 0.7780 | 27.36 | 0.7311 | 25.82 | 0.7812 | 29.77 | 0.9033 |
| | | GRAM [21] | 31.75 | 0.8913 | 28.33 | 0.7765 | 27.32 | 0.7302 | 25.62 | 0.7739 | 29.55 | 0.9003 |
| | | $\mathcal{L}_{\text{UDL}}$-Ours | **32.03** | **0.8949** | **28.60** | **0.7828** | **27.48** | **0.7355** | **26.21** | **0.7931** | **30.35** | **0.9097** |
| EDSR [2] | ×4 | Original | 32.17 | 0.8975 | 28.65 | 0.7856 | 27.59 | 0.7400 | 26.56 | 0.8043 | 30.66 | 0.9134 |
| | | GRAM [21] | 32.13 | 0.8963 | 28.57 | 0.7822 | 27.49 | 0.7362 | 26.19 | 0.7916 | 30.48 | 0.9097 |
| | | $\mathcal{L}_{\text{UDL}}$-Ours | **32.37** | **0.8986** | **27.74** | **0.7867** | **27.62** | **0.7407** | **26.65** | **0.8065** | **30.81** | **0.9149** |

loss functions such as *MSE* or $\mathcal{L}_1$ in terms of PSNR and SSIM values. Note that the improvements achieved by our proposed method do not bring any additional computation cost during testing time. Comparing EDSR-S ($1.5M$ parameters) with EDSR ($43M$ parameters), our proposed $\mathcal{L}_{\text{UDL}}$ can bring lightweight networks with more greater performance improvements than big ones. The visual image comparison results are reported in Fig. 4 and Fig. 5. As shown in Fig. 4, our proposed $\mathcal{L}_{\text{UDL}}$ has recovered with fewer visible artifacts (e.g., the circular pattern of the roof and the lines on the glassy surface) than original loss and GRAM [21]. Fig. 4 (f) depicts the uncertainty learned by our $\mathcal{L}_{\text{UDL}}$, revealing the challenging pixels with poor reconstruction performance. From Fig. 5, vertical center-line of window has been recover more clear with precisely estimated uncertainty shown in (e) and (f), while DPDNN and DPDNN-GRAM [21] failed to discern shown in (c) and (d) respectively. More visual comparisons can be found in supplementary material.

### 4.6 Results with BD Degradation Model

For blur downsampling (BD), we have compared proposed $\mathcal{L}_{\text{UDL}}$ loss function with GRAM [21] and original loss functions such as *MSE* or $\mathcal{L}_1$ on three different baseline networks. The average PSNR and SSIM results in Tab. 6 are retrained from officially released code. It is easy to see that our propose $\mathcal{L}_{\text{UDL}}$ loss function is superior to GRAM [21] and original loss functions such as *MSE* or $\mathcal{L}_1$ in terms of PSNR and SSIM values. The visual image comparison results of BD degradation are reported in Fig. 6 and Fig. 7. Note that the BD degradation involves Gaussian blur, increasing difficulty in recovering structure patterns. From Fig. 6, we can see that our SR result (Fig. 6 (e)) of 'Img 109' is the closest to that of the ground-truth. In another challenging image ('Img 078' from Urban100 [25]), our method can recover much more reliable textured details as shown in Fig. 7 (e); while all other methods have severe aliasing artifacts (i.e., distorted tile patterns). The visual quality improvement achieved by $\mathcal{L}_{\text{UDL}}$ is mainly due to the fact that our proposed method makes full use of the captured uncertainty to train deep networks focusing on the challenging pixels with high uncertainty. More visual comparisons can be found in our supplementary material.

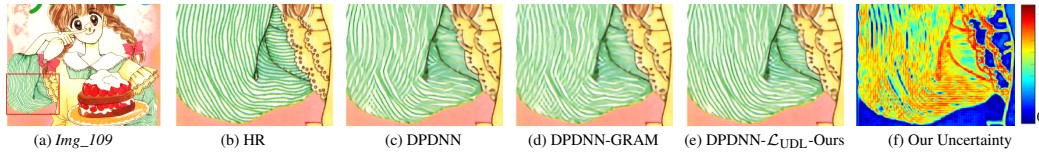

(a) *Img_109*  (b) HR  (c) DPDNN  (d) DPDNN-GRAM  (e) DPDNN-$\mathcal{L}_{\text{UDL}}$-Ours  (f) Our Uncertainty

Figure 6: SISR visual quality comparisons of DPDNN [3] with different loss function on 'Img_109' from Manga109 [26] (blur-downsampling $\times 4$). **Best viewed in color.**

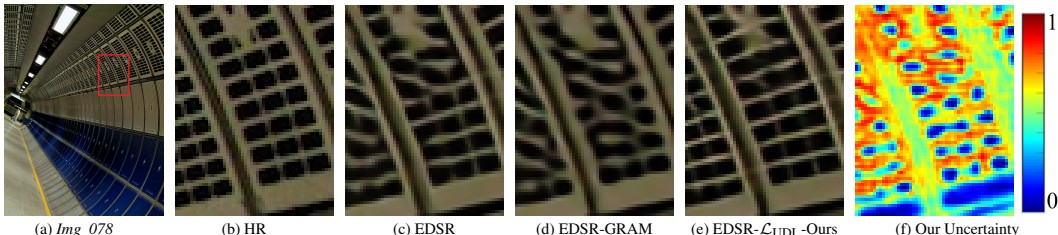

(a) *Img_078*  (b) HR  (c) EDSR  (d) EDSR-GRAM  (e) EDSR-$\mathcal{L}_{\text{UDL}}$-Ours  (f) Our Uncertainty

Figure 7: SISR visual quality comparisons of EDSR [2] with different loss function on 'Img_078' from Urban100 [25] (blur-downsampling $\times 4$). **Best viewed in color.**

## 5 Conclusion

In this paper, we propose a new adaptive weighted loss $\mathcal{L}_{\text{UDL}}$ for SISR to train SISR networks focusing on challenging pixels with high uncertainty (e.g., textured and edge pixels). Specifically, variance estimation is introduced into SISR so that the high-resolution images (mean) and their corresponding uncertainty (variance) can be learned simultaneously. Moreover, modeling uncertainty under Bayesian framework allows us to leverage sparsity prior for a more precise estimation of uncertainty. Ultimately, pixels with large certainty (e.g., texture and edge pixels) will be prioritized for SISR according to their importance to visual quality. For the first time, we demonstrate that such uncertainty-driven loss can achieve better results than *MSE* or $\mathcal{L}_1$ loss. Experimental results on three popular SISR networks show that our proposed uncertainty-driven loss has achieved better PSNR performance than traditional loss functions without any increased computation during testing.

## Acknowledgement

This work was supported in part by the National Key R&D Program of China under Grant 2018AAA0101400 and the Natural Science Foundation of China under Grant 61991451, Grant 61632019, Grant 61621005, and Grant 61836008. Xin Li's work is partially supported by the NSF under grants IIS-1951504 and OAC-1940855, the DoJ/NIJ under grant NIJ 2018-75-CX-0032, and the WV Higher Education Policy Commission Grant (HEPC.dsr.18.5).

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
