# Supplementary Material for
# Uncertainty-Driven Loss for Single Image Super-Resolution

**Qian Ning**[1], **Weisheng Dong**[1],[\*] **Xin Li**[2], **Jinjian Wu**[1], **Guangming Shi**[1]
[1]School of Artificial Intelligence, Xidian University, Xi'an 710071, China
[2]Lane Dep. of CSEE, West Virginia University, Morgantown WV 26506, USA
`ningqian@stu.xidian.edu.cn`, `{wsdong,jinjian.wu}@mail.xidian.edu.cn`
`xin.li@mail.wvu.edu`, `gmshi@xidian.edu.cn`

In this supplementary material, we analyse the influence of different scaling functions and training cost and we provide more visualization of the captured uncertainty on different networks and more visual comparison results on both bicubic downsampling (**BI**) and blur downsampling (**BD**) degradation.

## 1 Analysis of Different Scaling Functions

We have conducted experiments with twelve various monotonically increasing functions (including linear and non-linear) and the results can be summarized in Tab. 1.

Table 1: Average PSNR and $\Delta$ PSNR results with **BI** degradation on five datasets for investigating the influence of different scaling functions. The best and second-best performance are shown in **bold**.

| Scaling functions | Set5 | $\Delta$ | Set14 | $\Delta$ | BSD100 | $\Delta$ | Urban100 | $\Delta$ | Manga109 | $\Delta$ |
|---|---|---|---|---|---|---|---|---|---|---|
| Baseline | 31.61 | 0.00 | 28.22 | 0.00 | 27.30 | 0.00 | 25.25 | 0.00 | 29.31 | 0.00 |
| $s - min(s)$ | **31.90** | **0.29**↑ | **28.37** | **0.15**↑ | 27.40 | 0.10↑ | 25.54 | 0.29↑ | **29.77** | **0.46**↑ |
| $exp(s)$ | 31.80 | 0.19↑ | 28.34 | 0.12↑ | 27.40 | 0.10↑ | 25.53 | 0.28↑ | 29.66 | 0.35↑ |
| $exp(s)^2$ | 31.68 | 0.07↑ | 28.27 | 0.05↑ | 27.35 | 0.05↑ | 25.42 | 0.17↑ | 29.45 | 0.14↑ |
| $exp(s)^3$ | 31.15 | 0.46↓ | 27.98 | 0.24↓ | 27.19 | 0.11↓ | 25.06 | 0.19↓ | 28.82 | 0.49↓ |
| $exp(s)^{(1/2)}$ | 31.86 | 0.25↑ | 28.36 | 0.14↑ | **27.41** | **0.11**↑ | 25.55 | 0.30↑ | 29.71 | 0.40↑ |
| $exp(s)^{(1/3)}$ | 31.86 | 0.25↑ | 28.35 | 0.13↑ | 27.40 | 0.10↑ | 25.52 | 0.27↑ | 29.72 | 0.41↑ |
| $exp(s)^{(1/4)}$ | 31.86 | 0.25↑ | 28.35 | 0.13↑ | 27.40 | 0.10↑ | 25.51 | 0.26↑ | 29.72 | 0.41↑ |
| $log(s) - min(log(s))$ | **31.89** | **0.28**↑ | **28.39** | **0.17**↑ | 27.42 | 0.12↑ | **25.57** | **0.32**↑ | **29.74** | **0.43**↑ |
| $s^2$ | 31.85 | 0.24↑ | **28.37** | **0.15**↑ | 27.41 | 0.11↑ | 25.56 | 0.31↑ | 29.71 | 0.40↑ |
| $s^3$ | 31.80 | 0.19↑ | 28.33 | 0.11↑ | 27.40 | 0.10↑ | 25.53 | 0.28↑ | 29.65 | 0.34↑ |
| $s^{(1/2)}$ | 31.87 | 0.26↑ | **28.37** | **0.15**↑ | 27.40 | 0.10↑ | 25.54 | 0.29↑ | 29.72 | 0.41↑ |
| $s^{(1/3)}$ | 31.88 | 0.27↑ | 28.36 | 0.14↑ | 27.4 | 0.10↑ | 25.52 | 0.27↑ | 29.73 | 0.42↑ |

The best and second-best performances are shown in **bold**. Overall, twelve various monotonically increasing functions have achieved better results than the baseline method. The best two scaling functions are linear scaling and log scaling with a slight difference as shown in the above table. Since the linear scaling function achieves a comparable performance with low computational cost, we advocate this choice in this paper.

## 2 Analysis of Training cost

The training cost mainly depends on the original networks and iterative times. Let $n$ denotes the computational cost of forward inference of a batch of data. Then the computational cost of the backpropagation can be approximately equal to $n$. During our training experiments, the baseline

---

[\*]Corresponding author.

35th Conference on Neural Information Processing Systems (NeurIPS 2021).

method is trained with $4 \times 10^5$ minibatch updates. Step1 training process costs $1.2 \times 10^5$ minibatch updates to obtain a great uncertainty estimation. Step2 training process is trained with $4 \times 10^5$ minibatch updates and the variance branch doesn't require backpropagation. The computational complexity evaluation can be summarized in Tab. 2.

Table 2: The analysis of training cost of proposed method.

| - | Forward cost | Backpropagation cost | Iterative times | Computation cost |
|---|---|---|---|---|
| Baseline | $n$ | $n$ | $4 \times 10^5$ | $8 \times 10^5 n$ |
| Ours(step1) | $n$ | $n$ | $1.2 \times 10^5$ | $2.4 \times 10^n 5$ |
| Ours(step2) | $2n$ | $n$ | $4 \times 10^5$ | $1.2 \times 10^6 n$ |

Overall, the computational cost of the training process of our proposed method is 1.8 times of traditional MSE or L1 loss.

## 3  More visual comparison results of captured uncertainty.

Fig. 1 and Fig. 2 shows the uncertainty $\boldsymbol{\theta}$ captured by proposed $\mathcal{L}_{\text{UDL}}$ on three different SISR networks. From Fig. 1 and Fig. 2, we can see that the values of uncertainty $\boldsymbol{\theta}$ are consistent with the image edges and textures, revealing the challenging pixels which need to be prioritized since they are visually more important than smooth areas. With the aid from the estimated uncertainty $\boldsymbol{\theta}$, we propose uncertainty-driven loss $\mathcal{L}_{\text{UDL}}$ for SISR to train deep networks focusing on challenging situations such as textured and edge pixels with high uncertainty, achieving better performance than original loss function.

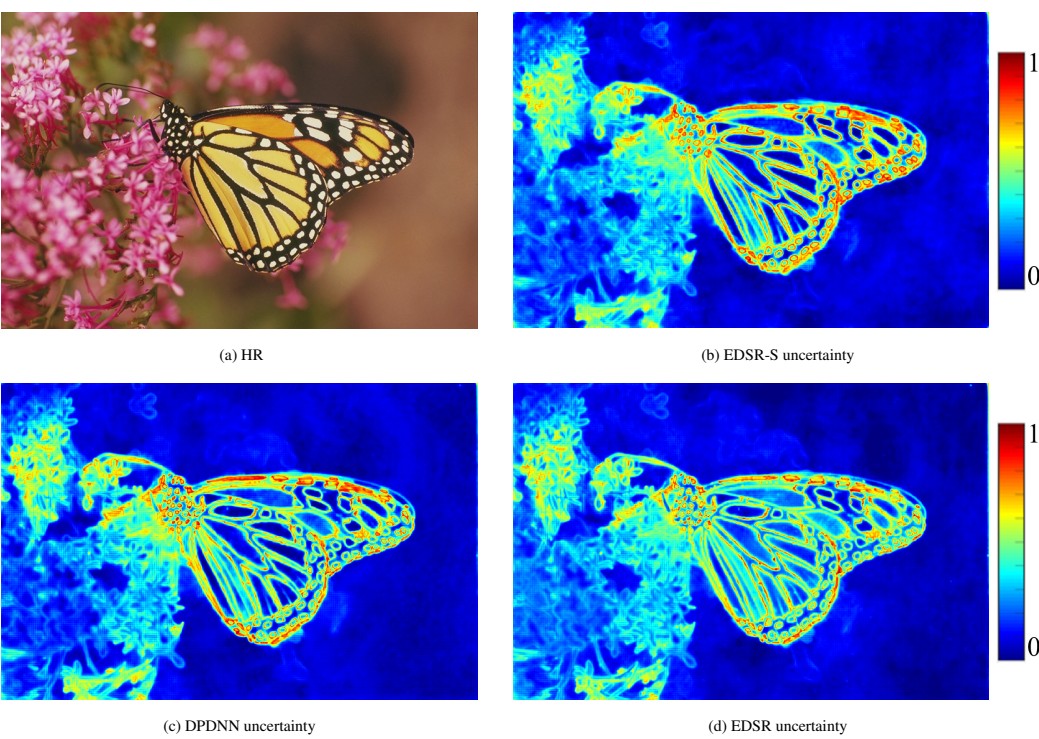

(a) HR

(b) EDSR-S uncertainty

(c) DPDNN uncertainty

(d) EDSR uncertainty

Figure 1: The uncertainty captured by proposed $\mathcal{L}_{\text{UDL}}$ loss with different SISR network on 'Img_011' from Set14 [1] (bicubic-downsampling $\times 3$).

## 4  More visual comparison results on BI degradation

The visual image comparison results on BI degradation are reported in Fig. 3, Fig. 4 and Fig. 5. From Fig. 3, we can see that our SR-resolved result of 'Img_046' from Urban100 is recovered with

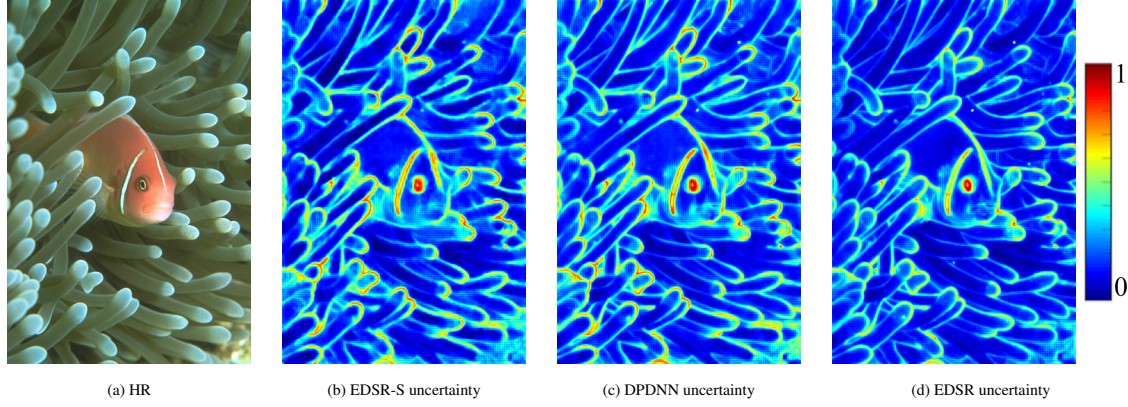

(a) HR  (b) EDSR-S uncertainty  (c) DPDNN uncertainty  (d) EDSR uncertainty

Figure 2: The uncertainty captured by proposed $\mathcal{L}_{\text{UDL}}$ loss with different SISR network on 'Img_040' from BSD100 [2] (bicubic-downsampling ×3).

fewer visible artifacts (e.g., the glassy surface of the building) and less blurry visual effect than other competing methods. In another challenging image ('Img 020' from Urban100 [3]), our method can recover much more reliable textured details as shown in Fig. 5 (d); while all other methods have severe aliasing artifacts (i.e., distorted texture of building). The visual quality improvement achieved by $\mathcal{L}_{\text{UDL}}$ can be credited to that our proposed method prioritize the challenging pixels with high uncertainty.

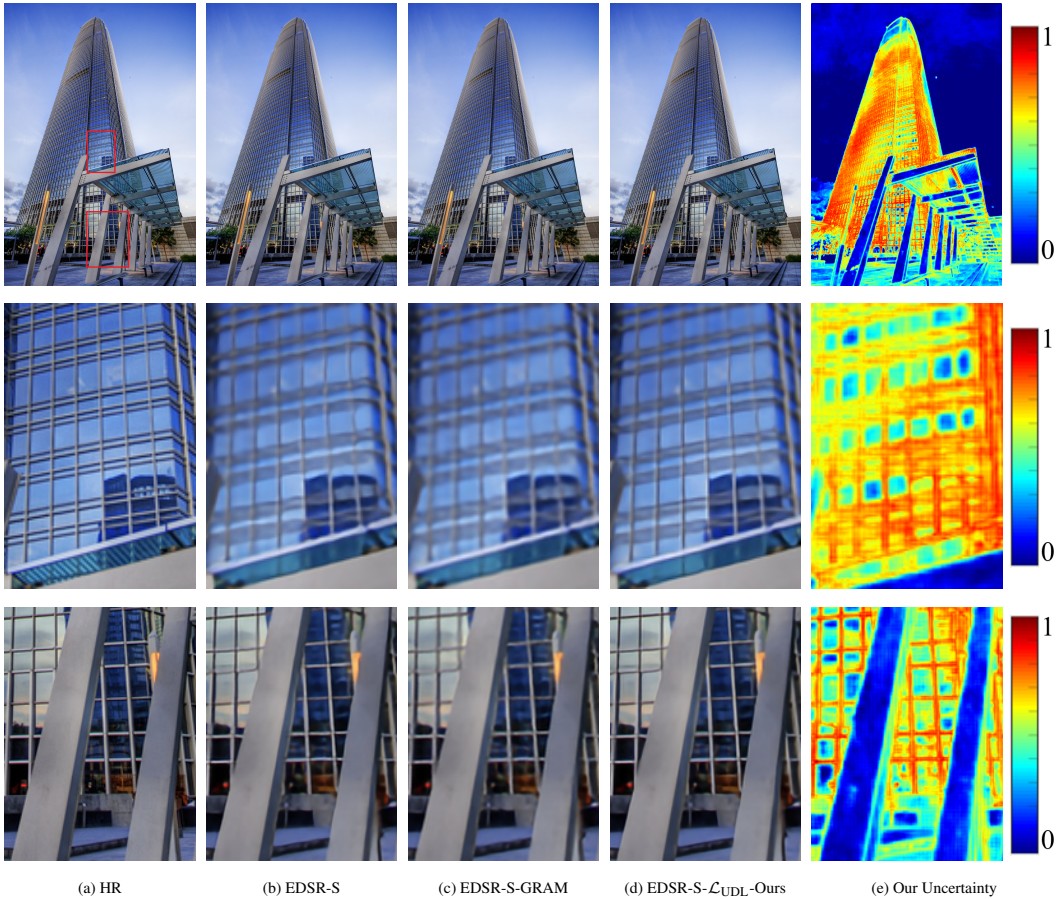

(a) HR  (b) EDSR-S  (c) EDSR-S-GRAM  (d) EDSR-S-$\mathcal{L}_{\text{UDL}}$-Ours  (e) Our Uncertainty

Figure 3: SISR visual quality comparisons of EDSR-S [4] with different loss function on 'Img_046' from Urban100 [3] (bicubic-downsampling ×3).

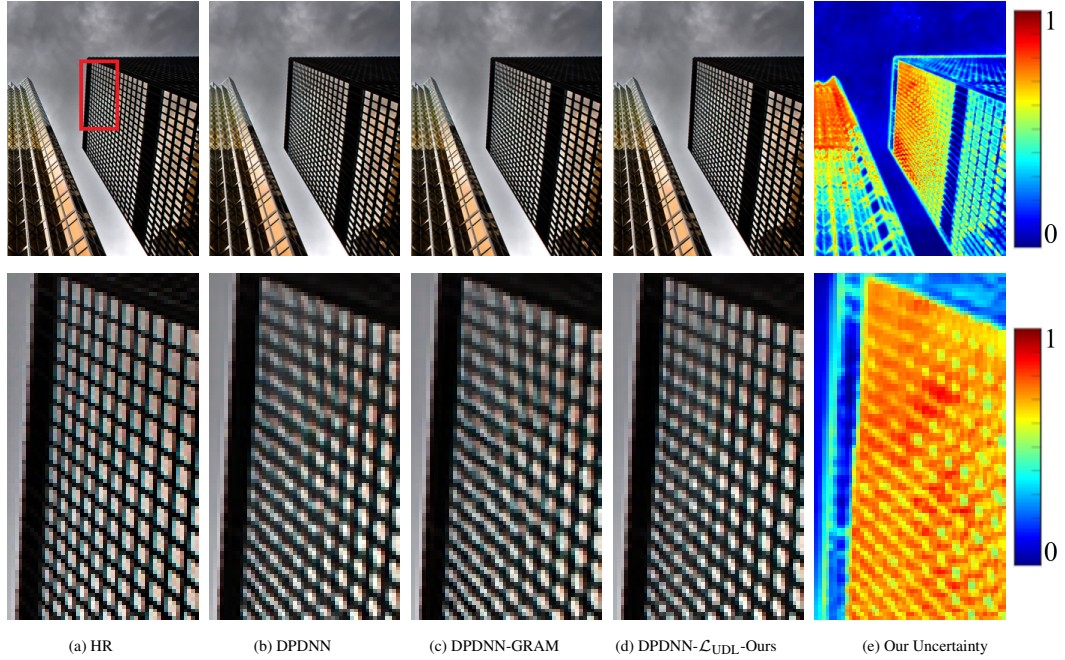

(a) HR  (b) DPDNN  (c) DPDNN-GRAM  (d) DPDNN-$\mathcal{L}_{\text{UDL}}$-Ours  (e) Our Uncertainty

Figure 4: SISR visual quality comparisons of DPDNN [5] with different loss function on 'Img_019' from Urban100 [3] (bicubic-downsampling $\times 2$).

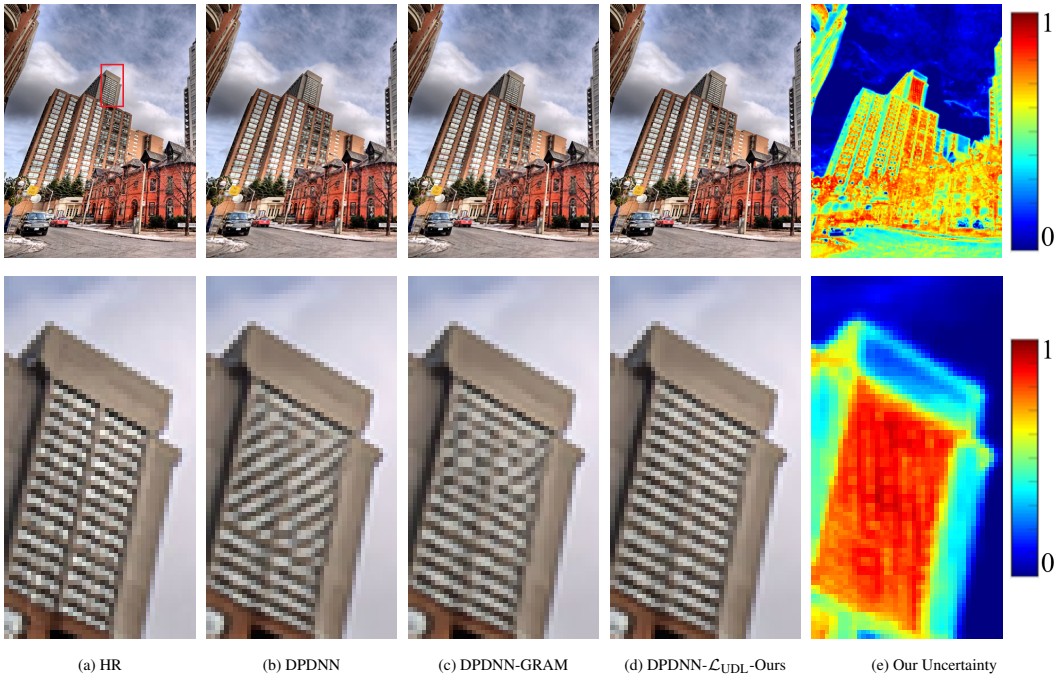

(a) HR  (b) DPDNN  (c) DPDNN-GRAM  (d) DPDNN-$\mathcal{L}_{\text{UDL}}$-Ours  (e) Our Uncertainty

Figure 5: SISR visual quality comparisons of DPDNN [5] with different loss function on 'Img_020' from Urban100 [3] (bicubic-downsampling $\times 2$).

# 5 More visual comparison results on BD degradation

The visual image comparison results on BD degradation with a scale factor of ×4 are reported in Fig. 6 and Fig. 7. Note that the BD degradation involves Gaussian blur, increasing difficulty in recovering structure patterns. From Fig. 6, we can see that our SR result (Fig. 6 (d) ) of 'Img_013' achieves the best visual results (i.e., the letter w). As shown in Fig. 7(d), subjective quality comparison results for a zebra image at the scale factor of ×4 are shown in Fig. 7. Apparently, our SR result of 'Img_052' from BSD100 [2]) is the closest to that of the ground-truth; while all other methods have severe aliasing artifacts (i.e., distorted patterns). The visual quality improvement achieved by $\mathcal{L}_{\mathrm{UDL}}$ can be credited to that our proposed method prioritize the challenging pixels with high uncertainty.

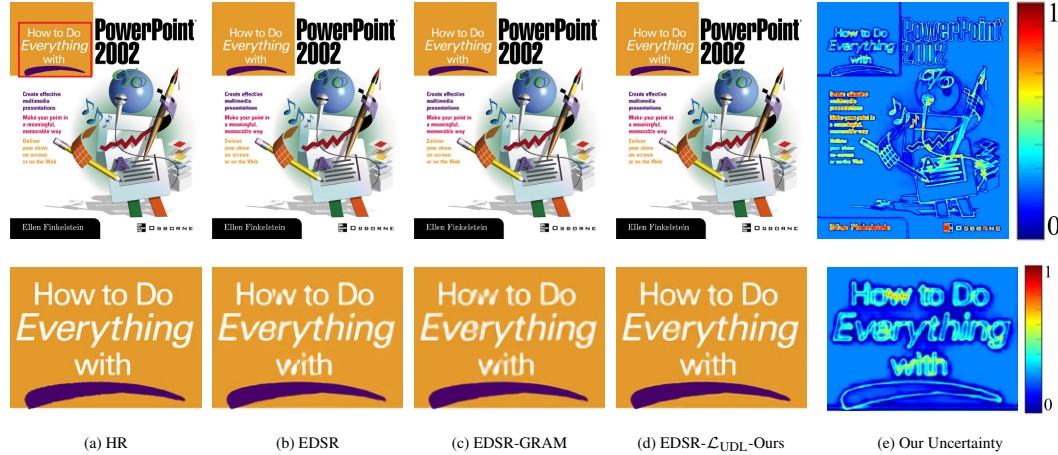

| (a) HR | (b) EDSR | (c) EDSR-GRAM | (d) EDSR-$\mathcal{L}_{\mathrm{UDL}}$-Ours | (e) Our Uncertainty |

Figure 6: SISR visual quality comparisons of EDSR [4] with different loss function on 'Img_013' from Set14 [1] (blur-downsampling ×4).

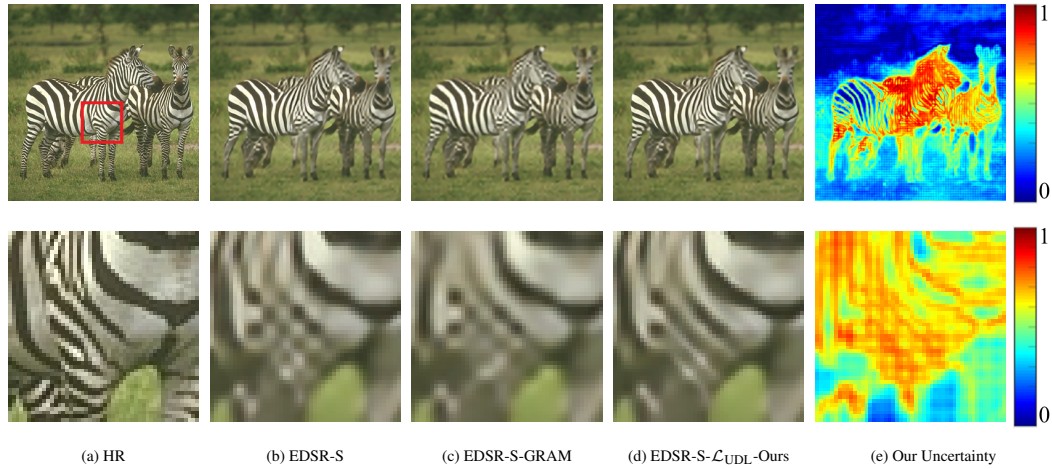

| (a) HR | (b) EDSR-S | (c) EDSR-S-GRAM | (d) EDSR-S-$\mathcal{L}_{\mathrm{UDL}}$-Ours | (e) Our Uncertainty |

Figure 7: SISR visual quality comparisons of EDSR-S [4] with different loss function on 'Img_052' from BSD100 [2] (blur-downsampling ×4).