# OpenReview forum: "Uncertainty-Driven Loss for Single Image Super-Resolution"
_NeurIPS.cc/2021/Conference — NeurIPS 2021 Poster_

### Official Review · Reviewer_oVep · 2021-07-15

**Rating:** 5
**Confidence:** 4

**Summary:**

This paper proposed a new loss for single image super-resolution by considering the uncertainty of each pixel within an image. Specifically, the authors claimed that pixels with larger uncertainty, such as texture and edge, should be more emphasized, and thus designed a loss function to achieve this goal. Experiments have been conducted to evaluated the effectiveness of the proposed method.

**Limitations And Societal Impact:**

The authors did not mention the limitations. As I can see, the major limitation of this method in practice is that it requires a two-step training, which increase the computational complexity.

**Main Review:**

The idea of emphasizing pixels with larger uncertainty is new, and seems effective. Experiments have also showed clear improvements as compared with baselines.

However, I do not think technical part this work is clearly clarified. Specifically,
1. There are many priors can induce sparsity other than Jeffrey's prior, and thus it should be justify why adopt this prior.

2. Also notice that, comparing Eq. (6) and Eq. (9), the difference only exists in the scalar multiplied to $s_i$. Therefore, it is natural to ask what the performance could be if we change this scalar to other values, which is also valid and can be derived similarly as Eq. (7) and Eq. (8), in the sense of generalize priors.

3. The final loss function in Eq. (10) is somewhat heuristic, since there are many ways to set $\hat{s}_i$ provided monotonically non-decreasing with respect to $s_i$.

Besides, many of the mathematics are not rigorous. For example,
1. In Eq. (3), the authors assume $\epsilon$ to be standard normal. However, according to equations after it, e.g., Eq. (4), $\epsilon$ should follow a Laplace distribution.

2. Eq. (7) is not correct, since $p(x_i|y_i,\theta_i)p(\theta_i)$ results in $p(x_i,\theta_i|y_i)$ according to conditional probability rule, but not $p(x_i|y_i)$.

3. Jeffrey's prior can only be written as $p(w)\propto\frac{1}{w}$, since it is not a valid distribution, and thus in Eq. (7) the equal sign after $p(x_i|y_i,\theta_i)p(\theta_i)$ should be replaced by "$\propto$".

**Time Spent Reviewing:**

6 hours.

---

> ### Author Response · Authors · 2021-08-10
> **Authors' Reply To Reviewer 4**
>
> We thank the reviewer for the time and effort on evaluating this work.
>
> **1." There are many priors can induce sparsity other than Jeffrey's prior, and thus it should be justified why adopt this prior. "**
>
> **Reply:** Jeffrey's prior is a strong and frequently used sparse prior in low-level vision tasks, such as [1-2], which has demonstrated its superiority. Therefore, we choose this prior empirically. Maybe Jeffrey's prior is not the best prior to inducing sparsity in SISR domain, but Jeffrey's prior server as a promising choice to explore the sparsity in uncertainty with improved results. Thanks for your suggestion, and we leave the task of finding the best sparse prior as our future work.
>
> [1] Wipf, David P., Bhaskar D. Rao, and Srikantan Nagarajan. "Latent variable Bayesian models for promoting sparsity." *IEEE Transactions on Information Theory* 57.9 (2011): 6236-6255.
>
> [2] Li Y, Dong W, Xie X, et al. Image super-resolution with parametric sparse model learning[J]. IEEE Transactions on Image Processing, 2018, 27(9): 4638-4650.
>
> **2."Also notice that, comparing Eq. (6) and Eq. (9), the difference only exists in the scalar multiplied to  $s_i$. Therefore, it is natural to ask what the performance could be if we change this scalar to other values, which is also valid and can be derived similarly as Eq. (7) and Eq. (8), in the sense of generalize priors. "**
>
> **Reply:** In fact, large values in this scalar won't lead to good performance, since it leads to too strong sparseness. In our experiments, the values $1$-$5$ would be appropriate choices. As the reviewer claims, we can adopt more generalize priors in order to obtain better results. In fact, we do think Jeffrey's prior is commonly used and generalized prior with very promising results. Thanks for your suggestion, and we leave finding the best sparse prior or more generalized priors in future work.
>
> **3."The final loss function in Eq. (10) is somewhat heuristic, since there are many ways to set $\hat s_i$ provided monotonically non-decreasing with respect to $s_i$."**
>
> **Reply:** Thanks for your constructive suggestions. We have conducted experiments with eleven various monotonically increasing functions, and the results can be summarized as follows:
>
> | Model  | Scale functions |   Set5    |   ΔPSNR   |   Set14   |   ΔPSNR   |  BSD100   |   ΔPSNR   | Urban100  |   ΔPSNR   | Manga109  |   ΔPSNR   |
> | :----: | :-------------: | :-------: | :-------: | :-------: | :-------: | :-------: | :-------: | :-------: | :-------: | :-------: | :-------: |
> | EDSR_S |    Baseline     |   31.61   |   0.00    |   28.22   |   0.00    |   27.30   |   0.00    |   25.25   |   0.00    |   29.31   |   0.00    |
> | EDSR_S |     s-min(s)      | **31.90** | **0.29↑** | **28.37** | **0.15↑** |   27.40   |   0.10↑   |   25.54   |   0.29↑   | **29.77** | **0.46↑** |
> | EDSR_S |     exp(s)      |   31.80   |   0.19↑   |   28.34   |   0.12↑   |   27.40   |   0.10↑   |   25.53   |   0.28↑   |   29.66   |   0.35↑   |
> | EDSR_S |    exp(s)^2     |   31.68   |   0.07↑   |   28.27   |   0.05↑   |   27.35   |   0.05↑   |   25.42   |   0.17↑   |   29.45   |   0.14↑   |
> | EDSR_S |  exp(s)^(1/2)   |   31.86   |   0.25↑   |   28.36   |   0.14↑   | **27.41** | **0.11↑** |   25.55   |   0.30↑   |   29.71   |   0.40↑   |
> | EDSR_S |  exp(s)^(1/3)   |   31.86   |   0.25↑   |   28.35   |   0.13↑   |   27.40   |   0.10↑   |   25.52   |   0.27↑   |   29.72   |   0.41↑   |
> | EDSR_S |  exp(s)^(1/4)   |   31.86   |   0.25↑   |   28.35   |   0.13↑   |   27.40   |   0.10↑   |   25.51   |   0.26↑   |   29.72   |   0.41↑   |
> | EDSR_S |log(s)-min(log(s))| **31.89** | **0.28↑** | **28.39** | **0.17↑** | **27.42** | **0.12↑** | **25.57** | **0.32↑** | **29.74** | **0.43↑** |
> | EDSR_S |       s^2       |   31.85   |   0.24↑   | **28.37** | **0.15↑** | **27.41** | **0.11↑** | **25.56** | **0.31↑** |   29.71   |   0.40↑   |
> | EDSR_S |       s^3       |   31.80   |   0.19↑   |   28.33   |   0.11↑   |   27.40   |   0.10↑   |   25.53   |   0.28↑   |   29.65   |   0.34↑   |
> | EDSR_S |     s^(1/2)     |   31.87   |   0.26↑   | **28.37** | **0.15↑** |   27.40   |   0.10↑   |   25.54   |   0.29↑   |   29.72   |   0.41↑   |
> | EDSR_S |     s^(1/3)     |   31.88   |   0.27↑   |   28.36   |   0.14↑   |   27.4    |   0.10↑   |   25.52   |   0.27↑   |   29.73   |   0.42↑   |
>
> The best and second-best performances are shown in **bold**. Overall, eleven various monotonically increasing functions have achieved better results than the baseline method. The best two scaling functions are linear scaling and log scaling with a slight difference as shown in the above table. Since the linear scaling function achieves a desirable performance, we advocate this choice. The analysis of various scaling functions will be added in the next revised version.
>
> Again, thank you for your constructive suggestions. We found them helpful and will include them in the next revised version.
>
> **4."the mathematics are not rigorous."**
>
> **Reply:** We thank the reviewer for pointing out those mathematics issues. we will correct those mathematics issues in the next revised version.
>
> **Authors' Final Comments** We hope that the discussion we provided alleviates the reviewer's concerns. We are happy to answer any additional questions the reviewer has.

---

### Official Review · Reviewer_WXXg · 2021-07-17

**Rating:** 5
**Confidence:** 3

**Summary:**

The authors proposed a new loss function guiding the SISR network focus more on the uncertain areas of the images. Specifically, the uncertainty map is obtained by the network trained with the proposed L_esu loss enforcing the sparsity in the map, and the final SISR network is initialized from the previous steps, and finetuned with the uncertainty-guided loss. The idea of using uncertainty guidance to train a SISR network will not increase the computational burden during inference, and the performance is demonstrated to be better than the baseline.

**Limitations And Societal Impact:**

See the above comments.

**Main Review:**

Pros,

- Interesting idea of estimating the uncertainty map and adding the sparsity prior.
- Better performance of using the guided training with the newly-proposed loss.

Cons,

- It demonstrates the uncertainty map is mostly focusing on edges and texture regions of the image. I wonder whether the proposed method estimating the uncertainly map will be equivalent to high-pass filtering or edge detector for image preprocessing. What if there are high-level noises existing in the input images, will it highly influence the uncertainty map?
- Given a general-purpose loss function potentially good for other image restoration tasks, I wonder what it performs on more SOTA models, and whether it will perform consistently better than the baseline. Current evaluation is not enough with only three models and two tasks. Since it seems fairly easy to plug the training framework into any models, why not run more experiments to make the model more solid?

My main concern is there can be multiple similar alternatives for the uncertainty estimation, and the authors had better clarify the motivations and compare the current way with some simpler methods. Besides, the evaluated models are not enough to show the effectiveness of the model. Some potential generalization ability is not shown in the limited experiments.

However, the simple loss function is intuitive and somehow brings some improvement on some specific models, I currently tend to accept the paper due to its strong intuition and clear presentation.


**Time Spent Reviewing:**

0.5

---

> ### Author Response · Authors · 2021-08-10
> **Authors' Reply To Reviewer 3**
>
> We thank the reviewer for the time and effort on evaluating this work.
>
> **1." It demonstrates the uncertainty map is mostly focusing on edges and texture regions of the image. I wonder  whether the proposed method of estimating the uncertainty map will be equivalent to high-pass filtering or edge detector for image preprocessing. What if there are high-level noises existing in the input images, will it highly   influence the uncertainty map? "**
>
> **Reply:** The physical meaning of uncertainty in our paper is the variances of recovered pixels, indicating the confidence of SR reconstruction, which is not related to high-pass filtering or edge detector for image preprocessing. In fact, we have conducted experiments in image denoising tasks with Gaussian noise. The uncertainty maps are still estimated accurately, despite the noise in the input images. The experimental results show consistent performance improvement over baseline across different datasets in image denoising task.
>
> **2."Given a general-purpose loss function potentially good for other image restoration tasks,  I wonder what it performs on more SOTA models, and whether it will perform consistently better than the baseline. Current evaluation is not enough with only three models and two tasks. Since it seems fairly easy to plug the training framework into any models, why not run more experiments to make the model more solid?"**
>
> **Reply:** Good point! Actually, we have experimented EDSR network with denoising tasks (achieving comparable performance with other SOTA methods) and the results show that our proposed loss function performs better than MSE loss. Note that we mainly focus on single image super-resolution in this paper. As for other image restoration tasks, we leave them as our future work.
>
> **3."My main concern is there can be multiple similar alternatives for the uncertainty estimation, and the authors had better clarify the motivations and compare the current way with some simpler methods.  "**
>
> **Reply:** Thanks for your constructive criticism. The physical meaning of uncertainty in our paper is the variances of recovered pixels, indicating the confidence of SR reconstruction. Despite error_map can be regraded as variance estimation of a single pixel, it is not accurate since the calculation of error_map is based on a single pixel.  Indeed, the pixel-wise gradient or edge map can show the edges of images, however, those maps also can't reflect the variance but only have a certain correlation to variance.  How to obtain an accurate estimation of the variance of recovered pixels is a challenging and unsolved problem. In this paper, we propose to estimate uncertainty (or called variance) by maximizing Laplace likelihood of pixels predictions from Bayes perspective.  Such approach gives a much more precise estimation of variance than other simple weighted maps such as *Error_map, HR_gradient_map and LR_gradient_map* . To better demonstrate the above claim, we have conducted experiments with a weighted loss function where the weight is a pixel-wise gradient or error map. The PSNR results of five benchmarks can be summarized as follows:
>
> | Model  |  Weighted loss  |   Set5    |   ΔPSNR   |   Set14   |   ΔPSNR   |  BSD100   |   ΔPSNR   | Urban100  |   ΔPSNR   | Manga109  |   ΔPSNR   |
> | :----: | :-------------: | :-------: | :-------: | :-------: | :-------: | :-------: | :-------: | :-------: | :-------: | :-------: | :-------: |
> | EDSR_S |    Baseline     |   31.61   |   0.00    |   28.22   |   0.00    |   27.30   |   0.00    |   25.25   |   0.00    |   29.31   |   0.00    |
> | EDSR_S |   Uncertainty   | **31.90** | **0.29↑** | **28.37** | **0.15↑** | **27.40** | **0.10↑** | **25.54** | **0.29↑** | **29.77** | **0.46↑** |
> | EDSR_S |    Error_map    |   31.77   |   0.16↑   |   28.30   |   0.08↑   |   27.35   |   0.05↑   |   25.40   |   0.15↑   |   29.57   |   0.26↑   |
> | EDSR_S | HR_gradient_map |   31.68   |   0.07↑   |   28.27   |   0.05↑   |   27.34   |   0.04↑   |   25.36   |   0.11↑   |   29.47   |   0.16↑   |
> | EDSR_S | LR_gradient_map |   31.69   |   0.08↑   |   28.29   |   0.07↑   |   27.35   |   0.05↑   |   25.38   |   0.13↑   |   29.50   |   0.19↑   |
>
> The best performance is shown in **bold**. The "HR_gradient_map" and "LR_gradient_map" denote calculating gradient map from high-resolution (ground truth) images and low-resolution images respectively. The calculation of gradient can be formulated as
>
>            V(i,j)= I(i+1,j)-I(i,j),   H(i,j)= I(i,j+1)-I(i,j), G(i,j)=||(V(i,j),H(i,j)||_2,
>
> where I denotes pixels value and i,j denotes position of pixels . Note that we adjust the scaling functions of *Error_map, HR_gradient_map and LR_gradient_map* to get the best performance.
>
> From the above table, one can be observed that other weighted loss functions can indeed improve the PSNR results, but only to certain degrees. Comparing four different weight maps, our proposed uncertainty weighted loss function can bring the biggest improvement. Although the error_map can represent the variance of a single pixel, the error_map lacks semantic information or local information to capture a more precise estimation of variance comparing uncertainty. With regard to the gradient map of HR or LR images, those gradient maps only well match the edges of images and have a certain correlation to variance.  Comparing the visual results of *Error_map, HR_gradient_map and LR_gradient_map* with *uncertainty map*, those maps only detect edges of images and fail reflecting complex texture details which are important to final reconstruction performance. Therefore, uncertainty-weighted loss can is still valuable for achieving the best performance among other weighted maps. To the best of our knowledge, there is no work except GRAM [20] in SISR domain that uses weighted loss to address the importance of different pixels, which is the biggest contribution and key novelty of our paper.
>
> Again, thank you for your constructive suggestions. We found them helpful and will include them in the next revised version.
>
> **Authors' Final Comments** We hope that the discussion we provided alleviates the reviewer's concerns. We are happy to answer any additional questions the reviewer has.

---

> > ### Comment · Reviewer_WXXg · 2021-08-31
> > **Thanks for the make-up exps**
> >
> > Thanks for the rebuttal, and my concerns are mostly addressed. I kind of like the idea of adding uncertainty to the loss function for learning image restoration task, and that's the intuition of this paper leads to my initial rating. However, according to other reviewers, this paper have some errors in formulations and some inaccurate presentation, and it seems going through another reviewing process will make this paper more solid. I will downgrade my rating a little.

---

### Official Review · Reviewer_H49w · 2021-07-20

**Rating:** 4
**Confidence:** 4

**Summary:**

This paper proposes a novel training scheme for end-to-end single image
super-resolution (SISR) with neural networks. The proposed scheme has two
steps. In the first step, a neural network is used to predict per-pixel
aleatoric uncertainty. In the second step, a second neural network is trained,
and the uncertainty estimated in the first step is used to weigh the per-pixel
restoration loss, so that uncertain pixels are more heavily penalized. Because
super-resolving textured regions and edges is typically more uncertain, the
authors claim that by making the model focus more on those regions, better
super-resolution results are obtained. Experimental results show consistent by
mild improvements over the baseline. In my opinion, the theoretical developments
need a stronger justification and their presentation needs to be improved.


**Limitations And Societal Impact:**

No.

**Main Review:**

Pros:
- Super-resolution is a one-to-many mapping so the outcome is inherently
  uncertain. It is good to see uncertainty being modeled in deep learning based
  SISR. This is a very recent trend that started with [21]. It also makes sense
  to me that the authors take into account the problem's heteorscedasticity
  since one expects uniform regions to exhibit less uncertainty in the output
  than texture or edges.
- Experimentally, the authors show  that their two-step training approach
  obtains a better perfoming model than the baseline (by 0.1 to 0.3db PSNR on
  average). Qualitative results in two examples show improvements over the
  baseline.


Cons:
- Since the goal of the work is to more strongly penalize textured and edge
  regions, a natural and simple baseline would be to optimize for the difference
  of image gradients (or other low-level fiters), or to weigh the penalization
  according to gradient magnitude. This would be much cheaper than having to
  recognize texture and edges with a secondary neural network.
- The theoretical developments in Section 3.1 have weak points that need to be
  clarified:
  - In lines 132, 137 authors say that they study only epistemic uncertainty,
    but it seems to me that what is being modeled is the aleatoric uncertainty:
    the formulation in (3) models heteroscedastic noise in the
    observations. Uncertainty of the model parameters (epistemic) does not
    appear in the presentation, to the best of my understanding.
  - In line 137 it is said that the observation noise in (3) is Gaussian but the
    likelihood equation (4) corresponds to Laplacian noise.
  - A Jeffrey's prior is introduced on one of the model's output (the estimated
    variance), by simply multiplying the likelihood function by one over the
    variance. I believe the probabilistic model intended here needs further
    clarification. Is the goal here to do a MAP formulation for this variance?
    Equation (7) is certainly confusing. The leftmost equation is invalid unless
    \theta_i is integrated over, and is independent of y_i. But then the
    probabilistic interpretation would not be clear, at least to me, since
    \theta_i is itself a function of y_i. Also, by multiplying the likelihood
    function by a factor it is no longer a probability over the x_i. I think all
    this deserves a better theoretical justification and an improved
    presentation.
  - There is no probabilistic justification for the second training step
    (weighing the reconstruction loss by the uncertainty).
- Although Figures 6 and 7 help to show the relative improvement of the proposed
  approach, it would help to include more examples and more state-of-the-art
  SISR methods in the comparison, so one can evaluate the validity of the
  approach as a general SISR method.

Minor:
- References [14] and [22] (Kendall and Gal 2017) are the same. This is a
  central reference to this work.



In my opinion, the weaknesses listed above outweight the submission's strengths.


**Time Spent Reviewing:**

5

---

> ### Author Response · Authors · 2021-08-10
> **Authors' Reply To Reviewer 2**
>
> We thank the reviewer for the time and effort on evaluating this work.
>
> **1." Since the goal of the work is to more strongly penalize textured and edge regions, a natural and simple baseline would be to optimize for the difference of image gradients (or other low-level fiters), or to weigh the penalization according to gradient magnitude. This would be much cheaper than having to recognize texture and edges with a secondary neural network."**
>
> **Reply:** Thanks for your constructive criticism. The physical meaning of uncertainty in our paper is the variances of recovered pixels, indicating the confidence of SR reconstruction. Despite error_map can be regraded as variance estimation of a single pixel, it is not accurate since the calculation of error_map is based on a single pixel.  Indeed, the pixel-wise gradient or edge map can show the edges of images, however, those maps also can't reflect the variance but only have a certain correlation to variance.  How to obtain an accurate estimation of the variance of recovered pixels is a challenging and unsolved problem. In this paper, we propose to estimate uncertainty (or called variance) by maximizing Laplace likelihood of pixels predictions from Bayes perspective.  Such approach gives a much more precise estimation of variance than other simple weighted maps such as *Error_map, HR_gradient_map and LR_gradient_map* . To better demonstrate the above claim, we have conducted experiments with a weighted loss function where the weight is a pixel-wise gradient or error map. The PSNR results of five benchmarks can be summarized as follows:
>
> | Model  |  Weighted loss  |   Set5    |   ΔPSNR   |   Set14   |   ΔPSNR   |  BSD100   |   ΔPSNR   | Urban100  |   ΔPSNR   | Manga109  |   ΔPSNR   |
> | :----: | :-------------: | :-------: | :-------: | :-------: | :-------: | :-------: | :-------: | :-------: | :-------: | :-------: | :-------: |
> | EDSR_S |    Baseline     |   31.61   |   0.00    |   28.22   |   0.00    |   27.30   |   0.00    |   25.25   |   0.00    |   29.31   |   0.00    |
> | EDSR_S |   Uncertainty   | **31.90** | **0.29↑** | **28.37** | **0.15↑** | **27.40** | **0.10↑** | **25.54** | **0.29↑** | **29.77** | **0.46↑** |
> | EDSR_S |    Error_map    |   31.77   |   0.16↑   |   28.30   |   0.08↑   |   27.35   |   0.05↑   |   25.40   |   0.15↑   |   29.57   |   0.26↑   |
> | EDSR_S | HR_gradient_map |   31.68   |   0.07↑   |   28.27   |   0.05↑   |   27.34   |   0.04↑   |   25.36   |   0.11↑   |   29.47   |   0.16↑   |
> | EDSR_S | LR_gradient_map |   31.69   |   0.08↑   |   28.29   |   0.07↑   |   27.35   |   0.05↑   |   25.38   |   0.13↑   |   29.50   |   0.19↑   |
>
> The best performance is shown in **bold**. The "HR_gradient_map" and "LR_gradient_map" denote calculating gradient map from high-resolution (ground truth) images and low-resolution images respectively. The calculation of gradient can be formulated as
>
>            V(i,j)= I(i+1,j)-I(i,j),   H(i,j)= I(i,j+1)-I(i,j), G(i,j)=||(V(i,j),H(i,j)||_2,
>
> where I denotes pixels value and i,j denotes position of pixels . Note that we adjust the scaling functions of *Error_map, HR_gradient_map and LR_gradient_map* to get the best performance.
>
> From the above table, one can be observed that other weighted loss functions can indeed improve the PSNR results, but only to certain degrees. Comparing four different weight maps, our proposed uncertainty weighted loss function can bring the biggest improvement. Although the error_map can represent the variance of a single pixel, the error_map lacks semantic information or local information to capture a more precise estimation of variance comparing uncertainty. With regard to the gradient map of HR or LR images, those gradient maps only well match the edges of images and have a certain correlation to variance.  Comparing the visual results of *Error_map, HR_gradient_map and LR_gradient_map* with *uncertainty map*, those maps only detect edges of images and fail reflecting complex texture details which are important to final reconstruction performance. Therefore, uncertainty-weighted loss can is still valuable for achieving the best performance among other weighted maps. To the best of our knowledge, there is no work except GRAM [20] in SISR domain that uses weighted loss to address the importance of different pixels, which is the biggest contribution and key novelty of our paper.
>
> Again, thank you for your constructive suggestions. We found them helpful and will include them in the next revised version.
>
> **2. "The theoretical developments in Section 3.1 have weak points that need to be clarified:"**
>
> **(1) In lines 132, 137 authors say that they study only epistemic uncertainty, but it seems to me that what is being modeled is the aleatoric uncertainty.**
>
> **(2) In line 137 it is said that the observation noise in (3) is Gaussian but the likelihood equation (4) corresponds to Laplacian noise.**
>
> **(3) The probabilistic model intended here needs further clarification.**
>
> **(4) There is no probabilistic justification for the second training step (weighing the reconstruction loss by the uncertainty).**
>
> **Reply:**
> (1) The uncertainty in our paper describes how much the model is uncertain about its predictions, indicating the difficulty of SR reconstruction. In [22], the epistemic/model uncertainty represents the uncertainty in model parameters and aleatoric/data uncertainty describes the noise in input data. We think there is a connection between model and data uncertainty, which can't be divided without any intersection. In this paper, we study the uncertainty that describes how much the model is uncertain about its predictions, which is related to not only model but also data. In this case, thanks for pointing it out, and we will change the corresponding terms in the next revised vision for a better clarify.
>
> (2) Thanks for pointing it out. It's a mistake, and we will correct this error in the next revised version.
>
> (3) Thanks for pointing it out. The mathematical rigor can be further improved, and we will correct this error in the next revised version.
>
> (4) The probabilistic justification for the second training step can be summarized as follows:
>
> The step1 loss is derived from below log likelihood:
>
> $ ~ ln~p(x_i,\theta_i|y_i) = -\frac{||x_i-f(y_i)||_1}{\theta_i} -ln(\theta_i)-ln2 $
>
> The step2 loss is derived from below log likelihood:
>
> $ ~ ln~p(x_i|\theta_i,y_i) = -\frac{||x_i-f(y_i)||_1}{ \hat \theta_i} -ln(\hat \theta_i)-ln2 \propto -\frac{||x_i-f(y_i)||_1}{ \hat \theta_i}$ where $\hat \theta_i = \frac{1}{log(\theta_i)}$ and $\theta_i$ is a constant given by step1.  During the step2 training process,  we treat $\hat \theta_i$ as a constant value given by step1. The probabilistic justification of step2 almost the same as step1 and the only difference is that the variance in step1 is a variate while the variance in step2 is a constant. Thanks for your suggestion, we will add above probabilistic justification in the next revised version.
>
> **3."Although Figures 6 and 7 help to show the relative improvement of the proposed approach, it would help to include more examples and more state-of-the-art SISR methods in the comparison, so one can evaluate the validity of the approach as a general SISR method. "**
>
> **Reply:** Due to limited pages, we have included some visual examples in our supplementary material. Thanks for your suggestion, we will add more examples and more state-of-the-art SISR methods in the comparison in the next revised version.
>
> **4."Minor error"**
>
> **Reply:** Thank you for pointing them out. We will correct this minor problem in the next revised version.
>
> **Authors' Final Comments** We hope that the discussion we provided alleviates the reviewer's concerns. We are happy to answer any additional questions the reviewer has.

---

> > ### Comment · Reviewer_H49w · 2021-08-31
> > **Thank you.**
> >
> > I thank the authors for thoroughly addressing my concerns and for their experimental results comparing to simple edge/texture detection.
> >
> > In my opinion the revisions to be made to the manuscript are too many at this point so I maintain my recommendation.

---

### Official Review · Reviewer_zyeM · 2021-07-22

**Rating:** 4
**Confidence:** 5

**Summary:**

This paper proposes a new loss function for the training of single image super-resolution. The proposed method is motivated by the uncertainty of predicting different pixels. Firstly, the uncertainty of each pixel in super-resolution is estimated. This uncertainty (variance) was used during training that the pixels with higher uncertainty were weighted with higher weights. A two-step training scheme was designed. The experiments show good results compared with other uncertainty-driven methods, such as GRAM.

**Ethics Review Area:**

["I don’t know"]

**Limitations And Societal Impact:**

Yes

**Main Review:**

Major concerns:

1. The novelty of this paper is somehow limited. The author describes a probability-based framework to explain the uncertainty in super-resolution. However, this part of the statement is common. Describe super-resolution under Bayesian statistics introduce no novelty. The Jeffrey's prior part is interesting but the author did not explain too much. The whole method is similar to training SR with weighted data (hard samples have higher weights). The relationship between the proposed method and weighted training is not clear (there is no expariments and statements about it).

2. In the experiment section, the author only presents comparison between the proposed method and GARM. More experiments are expected. Can we weight the pixel using other methods? such as calculate the gradient, high gradient indicates dense textures. Or weight the loss for each pixel based on Wevelet or DCT? Is there any difference in using different methods to weight loss? Why is the proposed weighting method better?

3. More explaination and experiments about the Jeffery's prior and its function in the proposed method is expected.

**Time Spent Reviewing:**

3

---

> ### Author Response · Authors · 2021-08-10
> **Authors' Reply To Reviewer 1**
>
> We thank the reviewer for the time and effort on evaluating this work.
>
> **1." The novelty of this paper is somehow limited. The author describes a probability-based framework to explain the uncertainty in super-resolution. However, this part of the statement is common. Describe super-resolution under Bayesian statistics introduce no novelty. The Jeffrey's prior part is interesting, but the author did not explain too much. The whole method is similar to training SR with weighted data (hard samples have higher weights). The relationship between the proposed method and weighted training is not clear (there is no experiments and statements about it). "**
>
> **Reply:** To the best of our knowledge, there is no work in SISR domain that using weighted loss to address the importance of different pixels, which is the biggest contribution and novelty of our paper. We are happy to read and compare the works that training SR with weighted data if the reviewer can provide. Additionally, we introduce uncertainty to describe the undesirable performance in pixels level from a probabilistic perspective, which we believe is novel in SR domain. The most related to our work is GRAM and we have discussed the difference in Sec 3.4. Note that our experimental results are dramatically better than those of GRAM.
>
> **2." In the experiment section, the author only presents comparison between the proposed method and GARM.  More experiments are expected. Can we weight the pixel using other methods? such as calculate the gradient,  high gradient indicates dense textures. Or weight the loss for each pixel based on Wevelet or DCT?   Is there any difference in using different methods to weight loss? Why is the proposed weighting method better? "**
>
> **Reply:** Thanks for your constructive criticism. The physical meaning of uncertainty in our paper is the variances of recovered pixels, indicating the confidence of SR reconstruction. Indeed, the pixel-wise gradient or edge map can show the edges of images, however, those maps can't reflect the variance but only have a certain correlation to variance. Still, the Wevelet or DCT maybe not be related to variance. How to obtain an accurate estimation of the variance of recovered pixels is a challenging and unsolved problem. In this paper, we propose to estimate uncertainty (or called variance) by maximizing Laplace likelihood of pixels predictions from Bayes perspective.  Such approach gives a much more precise estimation of variance than other simple weighted maps such as *Error_map, HR_gradient_map and LR_gradient_map* . To better demonstrate the above claim, we have conducted experiments with a weighted loss function where the weight is a pixel-wise gradient or error map. The PSNR results of five benchmarks can be summarized as follows:
>
> | Model  |  Weighted loss  |   Set5    |   ΔPSNR   |   Set14   |   ΔPSNR   |  BSD100   |   ΔPSNR   | Urban100  |   ΔPSNR   | Manga109  |   ΔPSNR   |
> | :----: | :-------------: | :-------: | :-------: | :-------: | :-------: | :-------: | :-------: | :-------: | :-------: | :-------: | :-------: |
> | EDSR_S |    Baseline     |   31.61   |   0.00    |   28.22   |   0.00    |   27.30   |   0.00    |   25.25   |   0.00    |   29.31   |   0.00    |
> | EDSR_S |   Uncertainty   | **31.90** | **0.29↑** | **28.37** | **0.15↑** | **27.40** | **0.10↑** | **25.54** | **0.29↑** | **29.77** | **0.46↑** |
> | EDSR_S |    Error_map    |   31.77   |   0.16↑   |   28.30   |   0.08↑   |   27.35   |   0.05↑   |   25.40   |   0.15↑   |   29.57   |   0.26↑   |
> | EDSR_S | HR_gradient_map |   31.68   |   0.07↑   |   28.27   |   0.05↑   |   27.34   |   0.04↑   |   25.36   |   0.11↑   |   29.47   |   0.16↑   |
> | EDSR_S | LR_gradient_map |   31.69   |   0.08↑   |   28.29   |   0.07↑   |   27.35   |   0.05↑   |   25.38   |   0.13↑   |   29.50   |   0.19↑   |
>
> The best performance is shown in **bold**. The "HR_gradient_map" and "LR_gradient_map" denote calculating gradient map from high-resolution (ground truth) images and low-resolution images respectively. The calculation of gradient can be formulated as
>
>            V(i,j)= I(i+1,j)-I(i,j),   H(i,j)= I(i,j+1)-I(i,j), G(i,j)=||(V(i,j),H(i,j)||_2,
>
> where I denotes pixels value and i,j denotes position of pixels . Note that we adjust the scaling functions of *Error_map, HR_gradient_map and LR_gradient_map* to get the best performance.
>
> From the above table, one can be observed that other weighted loss functions can indeed improve the PSNR results, but only to certain degrees. Comparing four different weight maps, our proposed uncertainty weighted loss function can bring the biggest improvement. Although the error_map can represent the variance of a single pixel, the error_map lacks semantic information or local information to capture a more precise estimation of variance comparing uncertainty. With regard to the gradient map of HR or LR images, those gradient maps only well match the edges of images and have a certain correlation to variance.  Comparing the visual results of *Error_map, HR_gradient_map and LR_gradient_map* with *uncertainty map*, those maps only detect edges of images and fail reflecting complex texture details which are important to final reconstruction performance. Therefore, uncertainty-weighted loss can is still valuable for achieving the best performance among other weighted maps. To the best of our knowledge, there is no work except GRAM [20] in SISR domain that uses weighted loss to address the importance of different pixels, which is the biggest contribution and key novelty of our paper.
>
> Again, thank you for your constructive suggestions. We found them helpful and will include them in the next revised version.
>
> **3." More explaination and experiments about the Jeffery's prior and its function in the proposed method is expected."**
>
> **Reply:** Jeffrey's prior is a very strong and frequently used sparse prior in many vision tasks, such as [1-2], which has demonstrated its superiority. Its function in the proposed method is to explore the sparsity of uncertainty since the uncertainty is sparse in view of the whole image as shown in Fig.2. We have explained the function of Jeffrey’s Prior in line 148 "Jeffrey’s Prior for Estimating Sparse Uncertainty (ESU) in SISR."
>
> [1] Wipf, David P., Bhaskar D. Rao, and Srikantan Nagarajan. "Latent variable Bayesian models for promoting sparsity." *IEEE Transactions on Information Theory* 57.9 (2011): 6236-6255.
>
> [2] Li Y, Dong W, Xie X, et al. Image super-resolution with parametric sparse model learning[J]. IEEE Transactions on Image Processing, 2018, 27(9): 4638-4650.
>
> **Authors' Final Comments** We hope that the discussion we provided alleviates the reviewer's concerns. We are happy to answer any additional questions the reviewer has.

---

> > ### Comment · Reviewer_zyeM · 2021-08-31
> > **Updated Review**
> >
> > After reading the author's rebuttal, I feel that my comment on novelty is a bit sloppy. Weighted loss is an interesting topic in this field, and few works really talk about it. I am looking forward to the profound insights that can be made. I am very happy that this paper is making some effort. But on the other hand, I think the technical novelty of this paper is relatively limited for NIPS. I will raise my rating to borderline.

---

### Decision · Program_Chairs · 2021-09-28

**Decision:**

Accept (Poster)

**Comment:**

This paper proposes a new weighted loss function for the training of single image super-resolution. The proposed scheme involves two steps. In the first step, the authors use a Bayesian estimation framework to predict mean and variance for each pixel in the high resolution image (not just a point estimate). Then the authors train a second neural network by weighting the reconstruction error according to the estimated variance. The intuition motivating this idea is that prediction errors on pixels with larger uncertainty (e.g. texture and edges) should be upweighted. Experimental evaluation shows that the new scheme is able to outperform the baseline.

The authors provided a constructive and high quality rebuttal, incorporating new experiments (including baselines and results using different monotonic functions to define the weighting scheme). However, all reviewers still consider the paper not ready to publish at NeurIPS.

Reviewers zyeM, oVep and H49w had concerns regarding the precision of the mathematical formulation. Specifically, several comments on the derivations in Section 3.1 and the lack of a probabilistic motivation for the second estimation step. This led Reviewer WXXg to reduce the score to 4, as she/he believes that the paper would benefit from a second revision. Reviewer H49w also expressed the same opinion.

Reviewer zyeM provided a score of 5 (updated from 4 after the rebuttal), although she/he did not change it in the system. While Reviewer zyeM considers the general direction of the work interesting (the investigation of weighting schemes for the SISR problem) and appreciated the detailed response provided by the authors, she/he still recommends rejecting the paper. The main reason is the lack of significant technical novelty in their approach.

Reviewers H49w and zyeM, suggested incorporating simple baselines that could identify the textured regions in the image without the need of training a full system. The authors added in their rebuttal convincing baselines, showing that the proposed approach can lead to singnigicantly better results.

All four reviewers recommend rejecting the work. The AC agrees with the concerns raised by the reviewers and finds no basis for overruling their recommendation.


**Consistency Experiment:**

NeurIPS has a long history of experimentation. In 2014, NeurIPS ran an experiment in which 10% of submissions were reviewed by two independent committees to quantify the randomness in the review process. This year, we repeated a variant of this experiment to see how the quality of the review process has changed over time.  This paper was part of the experiment and was therefore assigned to two committees (consisting of reviewers, an Area Chair, and a Senior Area Chair) that reached independent decisions.  If both committees made the same recommendation, this recommendation was followed. If a single committee recommended acceptance, the paper was accepted (with the exception of a few cases in which the other committee identified what we considered a fatal flaw, e.g., an error in a key result).

This copy’s committee reached the following decision: **Reject**

The other committee assigned to the paper recommended **Accept (Poster)**.  You can find the other set of reviews, along with any follow up discussion with the authors here:
https://openreview.net/forum?id=MXmmuhJYPdU